

# On the hydrostatic approximation in rotating stratified flow

Achim Wirth

Univ. Grenoble Alpes, CNRS, LEGI, F-38000 Grenoble, France

**Correspondence:** Achim Wirth (achim.wirth@legi.cnrs.fr)

**Abstract.** Hydrostatic models were and still are the workhorses for realistic simulations of the ocean dynamics, especially for climate applications. The hydrostatic approximation is formally first order in $\gamma = H/L$, where $H$ is the vertical and $L$ the horizontal scale of the phenomenon considered. For linear (low amplitude) and unforced stratified rotating flow the dynamics can be separated in balanced flow and wave motion. It is shown that for the linear balanced motion the hydrostatic approximation

is exact and for wave motion it is second order, obtaining the leading prefactors. The validity of the hydrostatic approximation therefore also relies on the ratio of the amplitude of wave motion to balanced motion. This ratio adds considerably to the quality of the hydrostatic approximation for larger scale flows in the atmosphere and the ocean.

Imposing the divergenceless condition is a linear projection of the dynamical variables into the subspace of divergenceless vector fields, for both the Navier-Stokes and the hydrostatic formalism. Both projections are local in Fourier space. The pro-

jection is followed by a time-evolution operator, which differs in the wave-frequencies, only. Combining the projection and the linear evolution operators in both formalisms leads to the linear projection-evolution operator.

Calculating the difference of the two projection-evolution operators, the expression of the error, scaling and prefactors, done by the hydrostatic approximation is obtained. Analyzing the eigen-space of the projector-evolution operators, it is shown that for rotating-buoyant vortical-flow the hydrostatic-approximation is of third order for buoyant forcing, second order for horizontal

and first order for vertical dynamical forcing. Equilibrium solutions are in the kernel of the linear projection-evolution operator and conservation laws are in the kernel of its adjoint.

Using the Heisenberg-Gabor limit it is shown that for large scale ocean dynamics, the difference of the dynamics of the projection-evolution operator between the two formalisms is insignificant. It is shown that the hydrostatic approximation is appropriate for realistic ocean simulations with vertical viscosities larger than $\approx 10^{-2} \text{m}^2\text{s}^{-1}$. A special emphasis is on unveiling

the physical interpretation of the calculations.

## 1   Introduction

At large scales the ocean velocities point predominantly in the horizontal direction, due to the flatness of the ocean basin, the action of gravity, the rotation of the earth and the feeble variations of density of ocean waters. This properties allow



for the derivation of idealized mathematical models of the ocean dynamics from the fully three dimensional Navier-Stokes equations (see, i.e. Vallis (2017)). Idealized models are not only implemented numerically to efficiently integrate atmosphere and ocean dynamics, but are at the basis of human understanding, Wirth (2010) and therefore of the developpement of story-lines, explaining physical processes and their interactions. Most of these models are based on the hydrostatic approximation. In the quasi-geostrophic model, for example, the entire dynamics is projected on the direction of the balanced eigen-vector

(to be defined below) and the influence of dynamics along the other vectors due to their linear and nonlinear interaction is estimated from the balanced flow. At small scales the turbulent motion is fully three-dimensional and the three-dimensional Navier-Stokes equations have to be solved. As numerical simulations, especially when coastal processes are considered, are moving towards finer horizontal resolution it is no-longer clear to what extent the hydrostatic approximation is valid. This has sparked an increased interest in quasi-hydrostatic or fully three-dimensional Navier-Stokes models for ocean modelling (see

i.e. Marshall et al. (1997a); Guillaume et al. (2017); Auclair et al. (2018); Popinet (2020); Calandrini et al. (2024)). At the same time hydrostatic models are still widely used and developed for larger scale simulations of the atmosphere (see e.g. Janjic et al. (2001); Wan et al. (2013); Milewski and Tabak (2015); Snodin and Wood (2022); Spensberger et al. (2022); Bouvier et al. (2024)), the ocean (see e.g. Kärnä et al. (2018); Kevlahan and Dubos (2019)) and the climate (see e.g. Boucher et al. (2020)). Hydrostatic models are also employed to increase physical insight (Atoufi et al. (2023); Winn et al. (2023)). Comparisons of

the two approaches using numerical simulations are reported (Tseng et al. (2005); Zeman et al. (2021)) as well as the coupling of the two model types, (Blayo and Rousseau (2016); Qu et al. (2019)). The present paper discusses the gray zone between the two limits and establishes, where it is situated, using a purely analytical approach.

   The justification of the hydrostatic approximation relies on the flatness of the dynamics, that is, the smallness of the ratio of the vertical scale to the horizontal scale, $\gamma = H/L$. It was recently proved by Li and Titi (2016) that the Navier–Stokes equations

converge strongly to the primitive (hydrostatic) equations and that the convergence is of first order, $O(\gamma)$. Scaling arguments on the validity of the hydrostatic approximation are often based on inertial-gravity wave dynamics. In Marshall et al. (1997b) it was estimated, comparing the horizontal advection time to the wave frequency, that the motion is hydrostatic if $\gamma^2/Ri \ll 1$, where $Ri = (NH/U)^2$ is the Richardson number, $N$ the buoyancy frequency and $U$ a typical horizontal velocity scale. These scaling arguments do not include the Coriolis parameter $f$, which is known to further suppress vertical accelerations, this can

be seen in the prominent Taylor-Proudman-Poincaré theorem (see i.e. Wirth and Barnier (2008)). Furthermore, the effect of resonances are not included. Note also that Gill (1982) states that the hydrostatic approximation is appropriate if the frequency of the wave motion is smaller than the buoyancy frequency $\omega \ll N$. For inertial-gravity waves $N > \omega > f$ and for balanced motion $\omega \ll N, f$. In the ocean and atmosphere we typically have $N > 10f$, but there are also areas, where the ocean and atmosphere are weakly or unstratified, or where the stratification is unstable. The action of gravity waves on the vortical flow

is, however, supposed to be small, rendering such arguments insignificant for most oceanic phenomena (see, i.e., Vanneste (2013), Wirth (2013)). In the present work I develop a frame work that allows for discussing the fidelity of the hydrostatic approximation and offers an understanding of the underlying physics. Calculations are detailed in the text as they unveil the physics. The calculations are performed in Fourier space, as projections into the space of incompressible vector fields are local, but results are by no means restricted spectral models.





The action of pressure, which insures the vanishing divergence of the velocity field, is discussed in Section 2. The present work combines three techniques: first (T1) I use the fact that imposing the divergenceless condition is a linear and local projection in Fourier space, for both, the Navier-Stokes and the hydrostatic formalism, it is discussed in Section 3. Second (T2), I extend the dynamical space to four dimensions considering the dynamical-vertical and buoyant acceleration as separate variables, it is discussed in Section 4. Third (T3) I use the linearized equations of the dynamics in both formalisms and compare

their eigen vectors and eigen values, to bring out the similarities and differences, it is discussed in Section 5. This leads to a formal frame-work to investigate the validity of the hydrostatic approximation using linear algebra in low ($\leq 5$) dimensional spaces.

To evaluate the hydrostatic approximation I consider the response of the linearized Navier-Stokes and the hydrostatic scheme to a forcing. Formally, the linear terms of the equations can be written on the l.h.s. of the equality sign and the other terms on the

r.h.s., the latter then appear, formally, as forcings to the linear system. The origin of these generalized forcing can be external or internal, by the boundary or by non-linear terms and other processes, which are explicitly resolved or parameterized. Two main steps are necessary to solve the two linearized schemes (see Fig. 2): The first step (S1) considers the effect of a forcing on the eigen-vectors of the corresponding linearized projection-evolution operator. This step is explained in Section 6. Section 7 details the different components of the forcing vector. The second step (S2) considers the response of the system to the forcing

in the eigen spaces, of the linear projection-evolution operator, which is given in Subsection 7.1 (S2a) for stationary forcing and in Subsection 7.2 (S2b) for time-dependent forcing. To evaluate the hydrostatic approximation, for each of the two steps, two sub-steps are necessary: one has to determine what is the error done by the hydrostatic approximation in each eigen-space, the absolute error, and what is the amplitude of the motion in each eigen-space, to obtain the relative error. The sub-steps are given in Subsection 7.1 for a stationary forcing and in Subsection 7.3 for a periodic forcing. In Section 7.4 I briefly discuss

the inclusion of the non-linear terms in the formalism. The discussing in Section 8 compares the error due to the hydrostatic approximation to other causes of error in numerical models as the increased friction coefficient and the Doppler shift due to unresolved advection. This section also discusses if the finite time observations of waves allows to discriminate between the two formalisms, based on the Heisenberg-Gabor uncertainty principle. The conclusions, Section 9, give a summary of the scaling due to different forcings and compares the deficiencies of the hydrostatic equations to unresolved processes, internal

variability, finite resolution in space and uncertainties in the observations.

### 1.1  The scales of ocean dynamics

Before evaluating the hydrostatic formalism by comparing its dynamics to the Navier-Stokes model, it is important to introduce the scales in space and time of the different regimes of ocean dynamics, as shown in Fig. 1. At small scales and short times the dynamics is three dimensional and the hydrostatic approximation deficient. We therefore consider the dynamics at scales

larger than 10 m and slower than 1 h. Four main regions can be identified. The first is the viscous region, below the blue line associated to the eddy viscosity employed in the model. The actual kinematic viscosity of sea water (O($10^{-6}$ m$^2$ s$^{-1}$)) is many orders of magnitudes smaller and the dynamics in the viscous region is misrepresented in both formalism. It is important to note that the vertical dynamics with typical vertical velocities of $w = 10^{-4}$ m s$^{-1}$ are in the viscous regime. The vertical





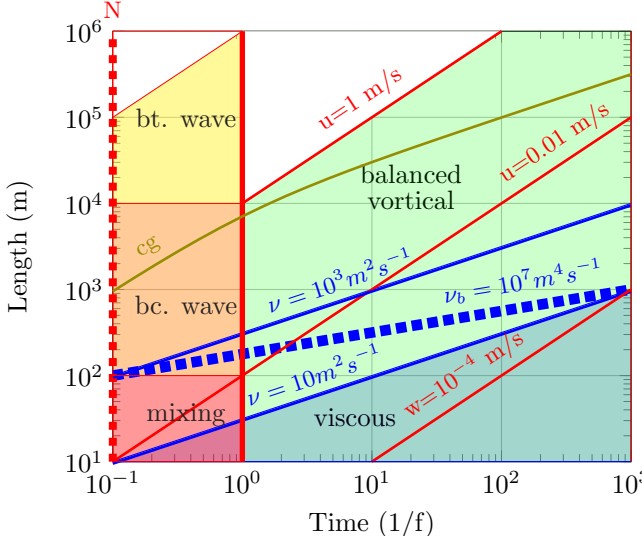

**Figure 1.** Schematic view of the ocean dynamics in (logarithmic) time and space scales. The time scale is normalized by the Coriolis period $1/f$, with $f = 10^{-4}\mathrm{s}^{-1}$. The buoyancy frequency is given by the red dashed line, a typical value is $N = 10f$. Inertia-gravity waves have periods shorter than $1/f$ and their group velocity $cg$ decreases with the wave length. At horizontal scales smaller than the depth of a homogeneous layer and not subject to substantial influence of the Coriolis force (red area), the dynamics can lead to overturning and mixing for which the hydrostatic approximation fails. Slow large-scale motion is close to balance (green area). Thin red-lines (slope=1) present constant velocity. Balanced motion in the ocean is rarely faster than 1 m s$^{-1}$, Barotropic gravity (bt.) waves (yellow area) have velocities of the order of 100 m s$^{-1}$. Internal gravity (bc.) waves (orange) are slower than 10 m s$^{-1}$ and typical vertical velocities are less than $10^{-4}$ m s$^{-1}$. The blue-lines (slope =1/2) represents the effect of viscous damping for different values of viscosity and also for bi-Laplace damping (blue-dashed-line, slope = 1/4). The area below is dominated by viscous friction, not correctly represented, neither in the hydrostatic nor the Navier-Stokes formalism.

velocities are crucial for the fidelity of the hydrostatic approxiametion, The second regime is where balanced vortical flow

dominates and the hydrostatic and the Navier-Stokes formalism are close, as confirmed by our findings below. This regime is operating at larger scales than the viscous regime and at scales faster than Coriolis parameter. At time scales faster than Coriolis parameter, the third regime representing inertia-gravity waves is dominant at larger scales and can be separated into barotropic and baroclinic modes. The former are much faster than the latter, both interact through topographic variations of the ocean floor. The former have negligible direct influence on the ocean circulation averaged over a few days. The latter can break

in the ocean interior leading to vertical mixing. The differences in the wave dynamics between both formalism is of second order in the aspect ratio $O(\gamma^2)$. The fourth regime is fully three dimensional, it is called the mixing regime in Fig. 1. Due to its three-dimensional dynamics the horizontal scale is constraint by the thickness of the ocean layer considered. In numerical simulations it is squeezed between the viscous and the wave regime and disappears for larger viscosities and hyper-viscosities. The validity of the hydrostatic approximation is questioned in this regime. It is, however, in this range of scales that a large



number of specific processes of ocean dynamics occur. Examples are convective plumes Marshall and Schott (1999); Wirth and Barnier (2006); Griffies and Treguier (2013), down-slope gravity currents Wirth (2009); Manucharyan et al. (2014); Gačić et al. (2021), interactions with topography Jayne et al. (2004), breaking of internal waves Lamb (2014), Ekman layers (top, bottom and interfacial) Laanaia et al. (2010); Elipot and Gille (2009), Langmuir cells McWilliams et al. (1997), ocean fronts McWilliams (2021), small scale instabilities Dewar et al. (2015) and others. Dedicated numerical simulations are necessary to

validate the hydrostatic approximation for these processes. In the present work I do not want to rival these efforts, but rather present a theoretical framework to consider its validity.

    In the ocean, the dynamics takes place also at scales faster than 1 h and smaller than 10 m. For this dynamics however neither rotation nor stratification play a dominant role, it is close to three-dimensional isotropic turbulence and standard large eddy simulation schemes developed in other fields of turbulent fluid dynamics can be employed to parameterize their effect on the

larger scale dynamics. For even smaller scale processes, as for example wave breaking and other surface processes, dedicated parameterizations have to be employed even in a far future.

## 2 Pressure

The starting point of our investigation are the three dimensional, incompressible Navier-Stokes equations in a rotating frame, subject to the Boussinesq approximation and the traditional approximation (see, i.e., Wirth and Barnier (2008)), which supposes

the rotation vector to be aligned with gravity:

$$\partial_t \mathbf{u} = -(\mathbf{u} \cdot \nabla)\mathbf{u} - f\mathbf{z} \times \mathbf{u} - g\mathbf{z}\frac{\rho}{\rho_0} + \nabla(\nu\nabla\mathbf{u}) - \nabla P, \tag{1}$$

$$\nabla \cdot \mathbf{u} = 0, \tag{2}$$

where $\mathbf{u}$ is the three dimensional velocity vector, $\mathbf{z}$ is the unit vector in the vertical upward direction, $g$ is gravity, $\nu$ viscosity and $P$ is the pressure divided by the average density (see below).

The density can be separated into three parts:

$$\rho(x, z, t) = \rho_0(t) + \rho_z(z, t) + \rho'(x, y, z, t). \tag{3}$$

(from left to right on the l.h.s. of Eq. 3): (1) the average density in the domain that can vary in time; (2) a part of the density variation that is horizontally averaged and only varies in the vertical direction and in time; (3) the deviations from the sum of the two. In the incompressible Navier-Stokes equations only the last term has to be considered for the acceleration, as the

other terms lead to a vertical force which is independent of the horizontal direction and therefore countered by a horizontally independent hydrostatic pressure force. We can replace $\rho$ by $\rho'$ and write the buoyancy $b = -g\rho'/\rho_0$ in Eq. 1 and the pressure will change from $P$ to $P'$, where the latter does not include the hydrostatic pressure force due to $\rho_0(t)$ and $\rho_z(z, t)$.

    If considering the terms on the r.h.s. of Eq. 1, from the left to the right, we see that the total acceleration is the sum of the advection of inertia $\mathbf{a}_u$ (first term on the r.h.s. of Eq. 1) , the Coriolis acceleration $\mathbf{a}_f$ (second term), buoyancy acceleration

$\mathbf{a}_b = (0, 0, b)^t$ (third term, the symbol $^t$ stands for the transpose of a vector). and the viscous acceleration $\mathbf{a}_\nu$ (fourth term).





The pressure term on the right hand side insures the vanishing divergence of the total acceleration, $\nabla \cdot \partial_t \mathbf{u} = 0$. Taking the divergence of Eq. 1 we can calculate the pressure by solving the elliptic equation:

$$\nabla^2 P = -\nabla \cdot (\mathbf{a}_u + \mathbf{a}_b + \mathbf{a}_f + \mathbf{a}_\nu), \tag{4}$$

subject to boundary conditions. Note that $\mathbf{a}_\nu$ does not appear in Eq. 4 if the viscosity is isotropic and constant in space, as the correct implementation of the viscous term is $\mathbf{a}_\nu = (\nabla^t \cdot \nu \nabla)\mathbf{u}$ and $\nabla \cdot \mathbf{u} = 0$ implies $\nabla \cdot \mathbf{a}_\nu = 0$.

If the flow is in geostrophic equilibrium (or more precisely, satisfies the thermal-wind relation) $\mathbf{a}_b = -\mathbf{a}_f$, $\mathbf{a}_u = \mathbf{a}_\nu = 0$ and there is no time evolution of the velocity field. The quasi-geostrophic dynamics is based on the hydrostatic balance. If a three dimensional perturbation is added to the dynamics it will approach quasi-geostrophy if rotation is a dominant feature of the dynamics. How fast and how close this tendency towards quasi-geostrophy happens, at different scales, is today an open question and depends on the problem considered (see, i.e., Vanneste (2013) and Wirth (2013)). Quasi-geostrophic flow is always close to balanced flow, which relies on the hydrostatic balance in the vertical and geostrophic balance in the horizontal. Viscosity perturbs the latter balance but not the former.

We can write the pressure as a sum of four parts, each insuring the zero divergence of the corresponding term on the r.h.s. of Eq. 10:

$$P(x,z,t) = -P_u(x,z,t) + P_f(x,z,t) + P_b(x,z,t) + P_\nu(x,z,t). \tag{5}$$

The first is related to the advection

$$\nabla^2 P_u = -\nabla \cdot ((\mathbf{u} \cdot \nabla)\mathbf{u}), \tag{6}$$

it is non-linear. The second is due to the Coriolis force:

$$\nabla^2 P_f = -\nabla \cdot (f\mathbf{z}\nabla \times \mathbf{u}). \tag{7}$$

Calculating the two pressures above asks for solving an elliptic (non-local) problem. The third term is due to buoyancy:

$$\partial_z P_b = b. \tag{8}$$

and can be solved by an integration in the vertical direction. If the viscosity is homogeneous and isotropic, the fourth term,

$$\nabla^2 P_\nu = \nabla \cdot (\nabla \cdot \nu \nabla \mathbf{u}), \tag{9}$$

vanishes.

The linear operator $\mathbb{P}$ projects the acceleration in the subspace of zero divergence (see Fig. 3). Equations 1 and 2 can therefore be written as:

$$\partial_t \mathbf{u} = -\mathbb{P}((\mathbf{u} \cdot \nabla)\mathbf{u}) - \mathbb{P}(f\mathbf{z} \times \mathbf{u}) + \mathbb{P}(\mathbf{z}b) - \mathbb{P}(\nu\nabla^2\mathbf{u}) \tag{10}$$

$$\mathbb{P}(\nabla \cdot \mathbf{u}) = 0 \tag{11}$$

Note, that although the projection is a linear operation, the pressure term is a non-linear term as it depends non-linearly on the velocity field through the non-linear advection term. Separating each step of a fluid dynamics integration in an evolution and





a projection sub-step is a classical procedure and is widely applied, also in ocean simulations (see discussion on projection in Marshall et al. (1997a); Guillaume et al. (2017); Popinet (2020)). In other numerical schemes Auclair et al. (2018) the evolution stays close to the subspace of divergenceless vector-fields by introducing a spurious compressibility.

Linear, elliptic, non-local, problems with constant coefficients become local algebraic problems in Fourier space and their

solution is reduced to solving linear algebraic systems. The orthogonal projection on the subspace of divergence-free functions is a local operation in Fourier space and it is therefore natural to present our formalism in Fourier space. The dilemma with the Fourier space is that boundary conditions, which are local in physical space become non-local in Fourier space, see Wirth (2005) for a detailed discussion and application to ocean modeling.

## 3 Fourier Space Projections (T1)

In this section I discuss the well known technique (labeled T1 in the introduction and in Fig. 2) of projections of a vector field into the subspace of divegenceless vector-fields for the Navier-Stokes formalism and develop the equivalent for the hydrostatic approximation. The Fourier transform of a $L^3-$periodic variable is:

$$\widehat{a}(\mathbf{k}) = \frac{1}{L^3} \int\limits_{L^3} e^{i(k_x x + k_y y + k_z z)} a(x, y, z) \, dx \, dy \, dz, \tag{12}$$

where the wave numbers are given by $k_x, k_y, k_z = 2\pi n/L$ and $n \in \mathbb{Z}$.

The integration starts from a divergence-free initial-condition $\widehat{\mathbf{u}}_i$ (see Fig. 3), with the integration step:

$$\widehat{a}_x = -ik_x \widehat{u_x^2} - ik_y \widehat{u_x u_y} - ik_z \widehat{u_x u_z} + f\widehat{u_y} - \tilde{\nu}\widehat{u_x} \tag{13}$$

$$\widehat{a}_y = -ik_x \widehat{u_x u_y} - ik_y \widehat{u_y^2} - ik_z \widehat{u_y u_z} - f\widehat{u_x} - \tilde{\nu}\widehat{u_y} \tag{14}$$

$$\widehat{a}_z = -ik_x \widehat{u_x u_z} - ik_y \widehat{u_y u_z} - ik_z \widehat{u_z^2} - \tilde{\nu}\widehat{u_z}, \tag{15}$$

$$\dot{\widehat{b}} = -ik_x \widehat{u_x b} - ik_y \widehat{u_y b} - ik_z \widehat{u_z b} - \widehat{u_z} N^2 - \kappa k^2 \widehat{b}, \tag{16}$$

with the wavelength $k = \sqrt{k_x^2 + k_y^2 + k_z^2}$, the inverse friction time $\tilde{\nu} = \nu k^2$ and the Brunt-Väisälä frequency $N = \sqrt{\partial_z b_{BG}}$ of the background stratification. The three-dimensional acceleration vector is:

$$\widehat{\mathbf{a}}^N = \begin{pmatrix} \widehat{a}_x \\ \widehat{a}_y \\ \widehat{a}_z + \widehat{b} \end{pmatrix}. \tag{17}$$

The hydrostatic pressure in Fourier space is $ik_z \widehat{P} = \widehat{b}$ and therefore $ik_x \widehat{P} = \frac{k_x}{k_z} \widehat{b}$ and the horizontal hydrostatic acceleration becomes: $-\frac{(k_x, k_y)^t}{k_z} \widehat{b}$. The two-dimensional acceleration in the hydrostatic formalism is:

$$\widehat{\mathbf{a}}^H = \begin{pmatrix} \widehat{a}_x - \frac{k_x}{k_z} \widehat{b} \\ \widehat{a}_y - \frac{k_y}{k_z} \widehat{b} \\ 0 \end{pmatrix}, \tag{18}$$




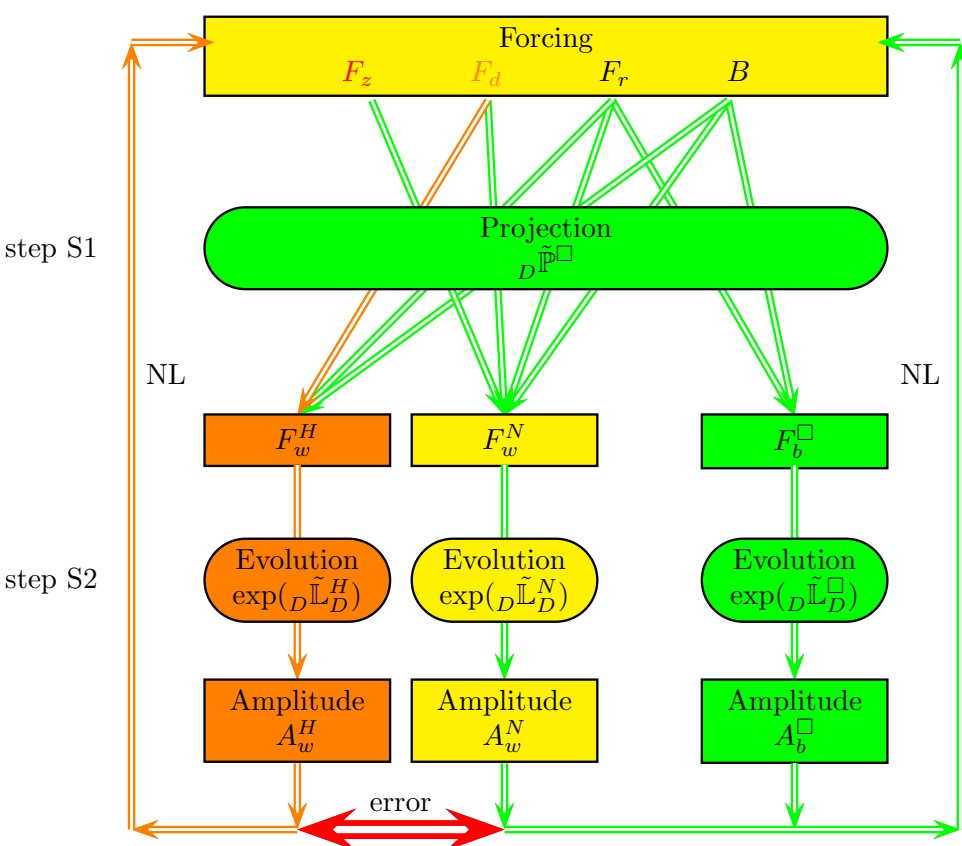

**Figure 2.** Schematic of the projection-evolution formalism: step S1 gives the difference in the projection of a forcing in the two formalisms and step S2 the difference in the linear evolution. Non-linear terms (NL) can be represented as (time-dependent) forcings and the scheme has to be iterated. The different forcings (vertical $F_z$, divergent $F_d$, rotational $F_r$ and buoyancy flux $B$) are projected on wave forcing $F_w$ and balanced forcing $F_b$, which span the sub-space of divergenceless vector-fields. The rotational and the buoyancy flux forcings are identical in both formalisms. The vertical forcing $F_z$ is in the kernel of the hydrostatic projection (missing arrow) and the error of the divergent forcing $F_d$ done by the hydrostatic projection is of second order $O(\gamma^2)$ (orange arrow). The projection-evolution operator acts identical on balanced forcing in both formalisms, whereas it differs to second order $O(\gamma^2)$ (orange arrow) on the wave forcing. The nonlinear terms formed by the wave and balanced amplitudes appear as retro-acting forces (NL) in the formalism.





that is, the hydrostatic pressure is added to the horizontal acceleration and the vertical acceleration is omitted.

The integration step is followed by a projection on the divergence-free sub-space. It is orthogonal to the divergence-free sub-space in the Navier-Stokes formalism,

$$\bar{\mathbb{P}}^N(\widehat{\mathbf{a}}^N) = \widehat{\mathbf{a}}^N - \frac{(\mathbf{k} \cdot \widehat{\mathbf{a}}^N)\mathbf{k}}{\mathbf{k}^2}. \tag{19}$$

This projection is local in Fourier space, it is performed independently for every wave vector $\mathbf{k}$ and makes the power of the pseudo-spectral method based on the Fourier series expansion of the velocity field. It was applied to ocean modelling in Wirth (2005).

For $k_z = 0$ the hydrostatic projection equals the Navier-Stokes projection applied to the first two components. For $k_z \neq 0$ the projection is in the vertical direction in the hydrostatic formalism leaving the horizontal components unchanged (see Fig.
200    3):

$$k_z = 0: \qquad \bar{\mathbb{P}}^H(\widehat{\mathbf{a}}^H) = \begin{pmatrix} 1 - \frac{k_x^2}{k_h^2} & -\frac{k_x k_y}{k_h^2} & 0 \\ -\frac{k_x k_y}{k_h^2} & 1 - \frac{k_x^2}{k_h^2} & 0 \\ 0 & 0 & 0 \end{pmatrix} \begin{pmatrix} \widehat{a}_x \\ \widehat{a}_y \\ 0 \end{pmatrix}, \tag{20}$$

$$k_z \neq 0: \qquad \bar{\mathbb{P}}^H(\widehat{\mathbf{a}}^H) = \begin{pmatrix} 1 & 0 & 0 \\ 0 & 1 & 0 \\ -\frac{k_x}{k_z} & -\frac{k_y}{k_z} & 0 \end{pmatrix} \widehat{\mathbf{a}}_H \tag{21}$$

with $k_h^2 = \sqrt{k_x^2 + k_y^2}$. It is clearly seen that the vertical acceleration prior to the projection is ignored, the entries of the third column vanish. The vertical acceleration is obtained, through the projection, entirely based on the horizontal components, such that the velocity field is divergenceless. Note that the correction for $k_z \neq 0$ are of order $\gamma = \sqrt{k_h^2/k_z^2}$, which is small for
dynamical processes which have a horizontal extension larger than the vertical. Both projectors lead to a divergenceless vector field, which is readily verified by multiplying the projectors by the wave-vector, $\mathbf{k}$, on their left.

The major difference between the hydrostatic and the Navier-Stokes projection is that in the latter the vertical acceleration from the dynamical and gravitational acceleration are treated equally, while in the former the vertical dynamical acceleration
is omitted and the gravitational acceleration is added to the horizontal acceleration using the hydrostatic equation, as shown in Fig. 3.

In numerical models based on Fourier representation the projection is exact at the machine precision employed. In other models the projection in the Navier-Stokes formalism is done via an elliptic solver of which the precision is specified and usually far less than the machine precision. In this case the double line in Fig. 3, representing the sub-space of zero divergence,
has a finite thickness corresponding to the precision prescribed to the elliptic solver.

## 4   Fourier Space Extension (T2)

In this section I introduce the extension of the Fourier space to 4 dimension, this technique is labeled T2 in the introduction and in Fig. 2. To emphasize the similarities and differences of both projections, I split the vertical acceleration into the two parts, a



dynamical ($\widehat{a}_z$) and a buoyant ($\widehat{a}_b = \widehat{b}$) part:

$$\tilde{\mathbf{a}} = \begin{pmatrix} \widehat{a}_x \\ \widehat{a}_y \\ \widehat{a}_z \\ \widehat{a}_b \end{pmatrix}. \tag{22}$$

The buoyancy adds to the dynamical vertical acceleration. In the Navier-Stokes formalism there is no difference in the projection of the two, while in the hydrostatic projection the dynamical vertical acceleration is ignored, while the buoyancy acceleration appears in the horizontal component. More precisely, the hydrostatic projection becomes:

$$\begin{aligned}
\mathbb{P}^H &= \begin{pmatrix} 1 & 0 & 0 & -\frac{k_x}{k_z} \\ 0 & 1 & 0 & -\frac{k_y}{k_z} \\ -\frac{k_x}{k_z} & -\frac{k_y}{k_z} & 0 & \frac{k_h^2}{k_z^2} \end{pmatrix} \\
&= \mathbb{1} - \frac{1}{k_z} \left[ \begin{pmatrix} 0 \\ 0 \\ 1 \end{pmatrix} (k_x, k_y, 0, -\frac{k_h^2}{k_z}) + \begin{pmatrix} k_x \\ k_y \\ 0 \end{pmatrix} (0,0,0,1) \right].
\end{aligned} \tag{23}$$

The third column is zero as the dynamical vertical acceleration is ignored, while it can be seen in the fourth column that the gravitational acceleration acts on the horizontal and vertical. The Navier-Stokes projection is:

$$\mathbb{P}^N = \begin{pmatrix} 1-\frac{k_x^2}{k^2} & -\frac{k_x k_y}{k^2} & -\frac{k_x k_z}{k^2} & -\frac{k_x k_z}{k^2} \\ -\frac{k_x k_y}{k^2} & 1-\frac{k_y^2}{k^2} & -\frac{k_y k_z}{k^2} & -\frac{k_y k_z}{k^2} \\ -\frac{k_x k_z}{k^2} & -\frac{k_y k_z}{k^2} & 1-\frac{k_z^2}{k^2} & 1-\frac{k_z^2}{k^2} \end{pmatrix} = \mathbb{1} - \frac{1}{k^2} \begin{pmatrix} k_x \\ k_y \\ k_z \end{pmatrix} (k_x, k_y, k_z, k_z). \tag{24}$$

The third and fourth column are identical, as the vertical acceleration due to the dynamical and gravitational part are treated equally. If buoyancy vanishes, the divergence free subspace is spanned by the orthogonal vectors $\mathbf{e}_1^P = (k_y, -k_x, 0, 0)^t$ and $\mathbf{e}_2^P = (k_x, k_y, -\frac{k_h^2}{k_z}, 0)^t$. There is no difference between the Navier-Stokes and the hydrostatic projections if applied to these two vectors. The first vector represents a divergenceless horizontal rotational motion with no vertical component. In two dimensions this is the only way to have a divergenceless mode. The second vector is a compensation between the horizontal and vertical divergence, it represents vertical stretching and compression of the water column. The vertical component of the curl of the second vector is vanishing.

To emphasize the projection on $\mathbf{e}_1^P$ and $\mathbf{e}_2^P$ the projection operators can also be written as:

$$\mathbb{P}^H = \frac{1}{k_h^2} \left[ \begin{pmatrix} k_y \\ -k_x \\ 0 \end{pmatrix} (k_y, -k_x, 0, 0) + \begin{pmatrix} k_x \\ k_y \\ -\frac{k_h^2}{k_z} \end{pmatrix} (k_x, k_y, 0, -\frac{k_h^2}{k_z}) \right] \tag{25}$$

$$\mathbb{P}^N = \frac{1}{k_h^2} \left[ \begin{pmatrix} k_y \\ -k_x \\ 0 \end{pmatrix} (k_y, -k_x, 0, 0) + \frac{k_z^2}{k^2} \begin{pmatrix} k_x \\ k_y \\ -\frac{k_h^2}{k_z} \end{pmatrix} (k_x, k_y, -\frac{k_h^2}{k_z}, -\frac{k_h^2}{k_z}) \right]. \tag{26}$$



There is no difference in the projection of the purely horizontal divergenceless motion, the first term in both projections.
Whereas, if horizontally divergent, buoyant and vertical motion is considered, there are two differences: the horizontally divergent and buoyant accelerations differ by a factor $k_z^2/k^2 = (1+\gamma^2)^{-1}$ and the vertical dynamic acceleration is neglected in the hydrostatic approximation. The former difference is of second order $O(\gamma^2)$, while the latter is of first order $O(\gamma)$, a higher order can be assigned if the dynamics constraints the order of the vertical dynamical acceleration $\widehat{a}_z$, this will be discussed in Section 5 based on the linearized, low amplitude, dynamics. Note that the projections are closely related to the well known Craya decomposition or wave-vortex compositions, see Deriaz et al. (2010) and references therein for a detailed discussion.

The vector $\mathbf{e}_3^P = (k_x, k_y, 0, k_z)^t$ is in the kernel of both projections and represents balanced flow: the hydrostatic balance in the vertical, $\widehat{a}_z = 0$, saying that all the acceleration in the vertical is due to buoyancy $\widehat{b}$, and the geostrophic balance in the horizontal: $\widehat{a}_x = f\widehat{u}_y = \frac{k_x}{k_z}\widehat{b}$ and $\widehat{a}_y = -f\widehat{u}_x = \frac{k_y}{k_z}\widehat{b}$. These compensations are the essence of Geophysical Fluid Dynamics (GFD), that is the dynamics of a buoyant fluid in a rotating frame. It includes geostrophy, thermal-wind balance and cyclostrophic balance.

The difference between the two projections lies in the second vector in the kernel, for the hydrostatic operator it formally is $\mathbf{e}_0^H = (0,0,1,0)$ but as, in the hydrostatic formalism, $\widehat{a}_z = 0$ it has zero amplitude. For the Navier-Stokes operator the second vector in the kernel is $\mathbf{e}_0^N = (0,0,1,-1)$, it expresses the trivial fact that compensations between the dynamical and gravitational accelerations are in the kernel of the operator. In the hydrostatic approximation ($\widehat{a}_z = 0$) no such compensation exists. Furthermore, $\mathbf{e}_{00}^N = k_z\mathbf{e}_0^N + \mathbf{e}_3^P = (k_x, k_y, k_z, 0)^t$ is also in the kernel of the Navier-Stokes operator, if multiplied by $\widehat{P'}_d$ it is nothing else than the dynamic pressure correction $\nabla P_d'$. Note that $\mathbf{e}_1^P, \mathbf{e}_2^P, \mathbf{e}_{00}^N$ are an orthogonal subset.

For $k_z \neq 0$ the difference between the operators is:

$$
\begin{aligned}
\mathbb{A}(\tilde{\mathbf{a}}) &= \mathbb{P}^H(\tilde{\mathbf{a}}) - \mathbb{P}^N(\tilde{\mathbf{a}}) = \\
&= \left[ \begin{pmatrix} 1 & 0 & 0 & -\frac{k_x}{k_z} \\ 0 & 1 & 0 & -\frac{k_y}{k_z} \\ -\frac{k_x}{k_z} & -\frac{k_y}{k_z} & 0 & \frac{k_h^2}{k_z^2} \end{pmatrix} - \begin{pmatrix} 1-\frac{k_x^2}{k^2} & -\frac{k_xk_y}{k^2} & -\frac{k_xk_z}{k^2} & -\frac{k_xk_z}{k^2} \\ -\frac{k_xk_y}{k^2} & 1-\frac{k_y^2}{k^2} & -\frac{k_yk_z}{k^2} & -\frac{k_yk_z}{k^2} \\ -\frac{k_xk_z}{k^2} & -\frac{k_yk_z}{k^2} & 1-\frac{k_z^2}{k^2} & 1-\frac{k_z^2}{k^2} \end{pmatrix} \right] \tilde{\mathbf{a}} \\
&= \frac{1}{k_z k^2} \underbrace{\begin{pmatrix} k_x^2 k_z & k_x k_y k_z & k_x k_z^2 & -k_x k_h^2 \\ k_x k_y k_z & k_y^2 k_z & k_y k_z^2 & -k_y k_h^2 \\ -k_x k_h^2 & -k_y k_h^2 & -k_z k_h^2 & (k_h^2)^2/k_z \end{pmatrix}}_{\mathbb{A}} \tilde{\mathbf{a}} \\
&= \frac{1}{k^2} \begin{pmatrix} k_x \\ k_y \\ -\frac{k_h^2}{k_z} \end{pmatrix} (k_x, k_y, k_z, -\frac{k_h^2}{k_z}) \tilde{\mathbf{a}}.
\end{aligned}
\tag{27}
$$

By construction, the difference of the two operators is in the space of divergenceless functions spanned by $\mathbf{e}_1^P$ and $\mathbf{e}_2^P$ (with the last component omitted), but the above shows that it is aligned with $\mathbf{e}_2^P$. This illustrates that the difference between the hydrostatic and the Navier-Stokes operator is never in the vertical vorticity, but in the compensation of horizontal divergence through vertical divergence, the water column is stretched differently if $(k_x, k_y, k_z, -\frac{k_h^2}{k_z})\tilde{\mathbf{a}} \neq 0$. The Navier-Stokes projector





does not lead to a change in vorticity, while a non-vanishing hydrostatic correction alters the horizontal-components of the vorticity vector, only.

If there is no constraint on the acceleration vector: $\sqrt{[\mathbb{A}(\tilde{\mathbf{a}})]^2} < \gamma \sqrt{\tilde{\mathbf{a}}^2}$, if $\gamma < 1$. Furthermore the error in the vertical is by a factor $\gamma$ smaller than the horizontal error. If the vertical dynamic acceleration is $\gamma$-times smaller than the horizontal or buoyant accelerations: $\sqrt{[\mathbb{A}(\tilde{\mathbf{a}})]^2} \leq \gamma^2 \sqrt{\tilde{\mathbf{a}}^2}$. Note that for the hydrostatic approximation to be of second order, $O(\gamma^2)$, only the vertical dynamic acceleration has to be small, not the buoyant. The relative error is $\mathbb{A}(\tilde{\mathbf{a}})/\mathbb{P}_N(\tilde{\mathbf{a}})$; for $\tilde{\mathbf{a}} = \mathbf{e}_0^N$ it is $\infty$.

So far, I have only considered the error in the hydrostatic approximation for an acceleration field, without any reference to the underlying dynamics. In the next section I will discuss the error for low amplitude motion, if the dynamics is well approximated by the linearized equations.

## 5   Linearized Equations (T3)

In this section I combine the projections in the extended Fourier space with the linearized evolution equation. If the velocity is small the first term on r.h.s. of Eq. 4 is negligible and the non-linear terms in Eqs. 13, 14, 15 and 16 can be neglected. When we further introduce a buoyancy velocity $u_b = b/N$ and write the four dimensional velocity vector $\hat{\mathbf{u}} = (u_x, u_y, u_z, u_b)^t$, the linearized dynamics is given by:

$$\begin{pmatrix} \tilde{\mathbf{a}} \\ \hat{u}_b \end{pmatrix} = \mathbb{L}^\square \hat{\mathbf{u}}, \tag{28}$$

where the symbol $\square$ is a place holder for " " for the Navier-Stokes or "H" for the hydrostatic formalism. Note that the vector on the r.h.s. has 5 dimensions as it consists of the accelerations was well as the buoyancy and the buoyancy flux divided by the Brunt-Väisälä frequency. This normalization renders the vectors and operators, introduced below, dimensionally homogeneous.

The extension to 5 dimensions is necessary to calculate the vertical acceleration from the dynamical and buoyant part and to keep track of the buoyancy anomaly, which is unchanged by the projection operator. The explicit form of the operators are:

$$\mathbb{L} = \begin{pmatrix} -\tilde{\nu} & f & 0 & 0 \\ -f & -\tilde{\nu} & 0 & 0 \\ 0 & 0 & -\tilde{\nu} & 0 \\ 0 & 0 & 0 & N \\ 0 & 0 & -N & -\kappa k^2 \end{pmatrix} \quad \text{and}$$

$$\mathbb{L}^H = \begin{pmatrix} -\tilde{\nu} & f & 0 & 0 \\ -f & -\tilde{\nu} & 0 & 0 \\ 0 & 0 & -\tilde{\nu} & 0 \\ 0 & 0 & 0 & N \\ N\frac{k_x}{k_z} & N\frac{k_y}{k_z} & 0 & -\kappa k^2 \end{pmatrix}. \tag{29}$$

I used $\partial_t b = -u_z N^2 - \kappa k^2 b$.





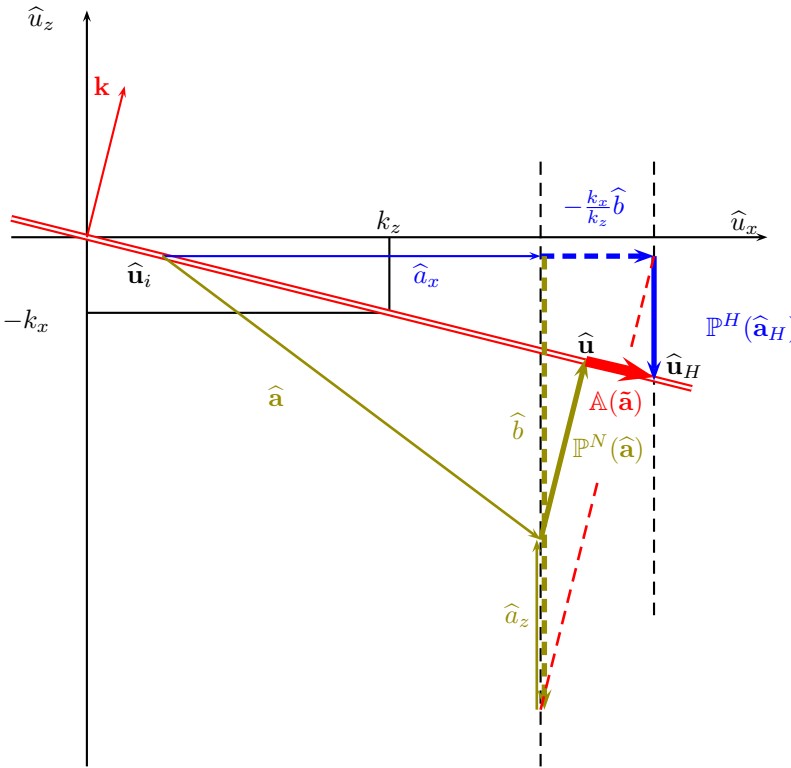

**Figure 3.** Schematic view of the (real-component) Fourier subspace spanned by the velocity components $(\widehat{u}_x(k_x, k_z), \widehat{u}_z(k_x, k_z))$ for the mode $\mathbf{k} = (k_x, k_z \neq 0)$, given by the thin-red-vector. The corresponding perpendicular subspace of zero divergence is shown by the double-red-line. Points are velocities and vectors are accelerations multiplied by the time step $\Delta t$. One time-step in the hydrostatic (blue) and Navier-Stokes (olive) formalism is shown. It consists in an evolution followed by a projection step. The evolution step (thin-full-vector) $\partial_t \widehat{\mathbf{u}}$ starts from the initial velocity $\widehat{u}_i$. In the hydrostatic formalism the vertical acceleration is neglected during the evolution step. The buoyancy acceleration in the Navier-Stokes formalism is given by the dashed olive vector. When added to the dynamic vertical acceleration $a_z$ it gives the total vertical acceleration. The buoyancy correction in the hydrostatic formalism, is given by the blue dashed line, it is in the horizontal and added to the horizontal acceleration . It is obtained by projecting the buoyancy acceleration along the wave vector (dashed-red-line) on the line defined by the horizontal acceleration. The projection step (thick-full-vectors) in both formalisms end on the double-red-line, representing vanishing divergence. The Navier-Stokes projection is orthogonal to the subspace of zero-divergence and ends at $\widehat{\mathbf{u}}$. The hydrostatic projection is in the $\widehat{u}_z$ direction and ends at $\widehat{\mathbf{u}}_H$. The difference between the hydrostatic and the Navier-Stokes formalism, the error, is given by the thick-red-vector. The dashed red-line is parallel to the wave-vector.



We define the extended projection operator:

$$\tilde{\mathbb{P}}^\square = \begin{pmatrix} \mathbb{P}^\square & 0 \\ 0 & 1 \end{pmatrix}. \tag{30}$$

The change in buoyancy is unaffected by the divergenceless condition, but the acceleration has to be projected into the subspace of divergence-free vector-fields. The operator $\tilde{\mathbb{L}}^\square = \tilde{\mathbb{P}}^\square \mathbb{L}^\square$ governs the time evolution via:

$$\partial_t \widehat{\mathbf{u}}^\square(t) = \tilde{\mathbb{L}}^\square \widehat{\mathbf{u}}^\square(t) + \tilde{\mathbb{P}}^\square \widehat{\mathbf{F}}(t). \tag{31}$$

The force $\widehat{\mathbf{F}} = (\widehat{F}_x, \widehat{F}_y, \widehat{F}_z, 0, \widehat{B})^t$ is formed of forces in the first three components. The fourth component is vanishing as there is no buoyancy imposed by the exterior but a buoyancy flux which is in the fifth component, when divided by the the Brunt-Väisälä frequency. The forces and fluxes are divided by the reference density and represent accelerations and relative fluxes, in the Boussinesq approximation used here. The explicit forms of the evolution operators are:

$$\tilde{\mathbb{L}} = \begin{pmatrix} -\nu(k^2 - k_x^2) + f\frac{k_x k_y}{k^2} & \nu k_x k_y + f\frac{k^2 - k_x^2}{k^2} & \nu k_x k_z & -N\frac{k_x k_z}{k^2} \\ \nu k_x k_y - f\frac{k^2 - k_y^2}{k^2} & -\nu(k^2 - k_y^2) - f\frac{k_x k_y}{k^2} & \nu k_y k_z & -N\frac{k_y k_z}{k^2} \\ \nu k_x k_z + f\frac{k_y k_z}{k^2} & \nu k_y k_z - f\frac{k_x k_z}{k^2} & -\nu k_h^2 & N\frac{k_h^2}{k^2} \\ 0 & 0 & -N & -\kappa k^2 \end{pmatrix} \tag{32}$$

$$\tilde{\mathbb{L}}^H = \begin{pmatrix} -\tilde{\nu} & f & 0 & -N\frac{k_x}{k_z} \\ -f & -\tilde{\nu} & 0 & -N\frac{k_y}{k_z} \\ \tilde{\nu}\frac{k_x}{k_z} + f\frac{k_y}{k_z} & \tilde{\nu}\frac{k_y}{k_z} - f\frac{k_x}{k_z} & 0 & N\frac{k_h^2}{k_z^2} \\ N\frac{k_x}{k_z} & N\frac{k_y}{k_z} & 0 & -\kappa k^2 \end{pmatrix} \tag{33}$$

If diffusivity equals viscosity, $\kappa = \nu > 0$, both matrices have two complex conjugated eigen values $\lambda_{1,2}^\square = \pm i\omega^\square - \tilde{\nu}$, where:

$$\omega^2 = \frac{\alpha}{k^2}, \qquad (\omega^H)^2 = \frac{\alpha}{k_z^2} \qquad \text{with} \qquad \alpha = f^2 k_z^2 + N^2 k_h^2, \tag{34}$$

are the dispersion relations of internal waves in the Navier-Stokes and the hydrostatic formalism, respectively. They are equal if and only if the horizontal wave numbers vanish, $k_h = 0$. Note that for the Navier-Stokes formalism the frequency of the waves, $\omega$, is between the Coriolis $f$ and the buoyancy frequency $N$, whereas for hydrostatic waves it, $\omega^H$, is above the Coriolis frequency. For $k_h^2 > k_z^2$ the hydrostatic approximation allows for waves with square frequencies faster than the maximum of $f^2$ and $N^2$. I use the superscript $\square$ when either formalism is concerned. The other two eigen-values are $\lambda_3 = -\tilde{\nu}$ and $\lambda_4 = 0$.



The corresponding eigen vectors are:

$$
\mathbf{e}^{\square}_{1,2} = \begin{pmatrix} \pm i\tilde{\gamma}^{\square}\omega\frac{k_x}{k_h} + f\frac{k_y}{k_h} \\ \pm i\tilde{\gamma}^{\square}\omega\frac{k_y}{k_h} - f\frac{k_x}{k_h} \\ \mp \frac{i\tilde{\gamma}^{\square}\omega k_h}{k_z} \\ \frac{Nk_h}{k_z} \end{pmatrix}, \quad \mathbf{e}_3 = \frac{f}{k_h}\begin{pmatrix} k_y \\ -k_x \\ 0 \\ -\frac{f}{N}k_z \end{pmatrix},
$$

$$
\mathbf{e}_4 = \frac{1}{f^2 k_h}\begin{pmatrix} (\tilde{\nu}^2 + N^2)[fk_y + \tilde{\nu}k_x] \\ (\tilde{\nu}^2 + N^2)[-fk_x + \tilde{\nu}k_y] \\ \tilde{\nu}(f^2 + \tilde{\nu}^2)k_z \\ -N(f^2 + \tilde{\nu}^2)k_z \end{pmatrix}, \quad \mathbf{e}^H_4 = \begin{pmatrix} 0 \\ 0 \\ f \\ 0 \end{pmatrix}, \tag{35}
$$

The first two vectors present inertial oscillations which are fast motion. We see that the eigen vectors for the fast motion and the corresponding eigen values differ by $\omega^H = \tilde{\gamma}^H\omega$, where we set $\tilde{\gamma}^H = \sqrt{1+\gamma^2}$, This shows that the error is of second order, $O(\gamma^2)$. The first two, complex valued vectors, can be substituted by the real orthogonal vectors,

$$
\tilde{\mathbf{e}}_1 = \begin{pmatrix} f\frac{k_y}{k_h} \\ -f\frac{k_x}{k_h} \\ 0 \\ \frac{Nk_h}{k_z} \end{pmatrix}, \quad \tilde{\mathbf{e}}_2 = \omega\begin{pmatrix} \frac{k_x}{k_h} \\ \frac{k_y}{k_h} \\ -\frac{k_h}{k_z} \\ 0 \end{pmatrix}, \tag{36}
$$

of which the directions are independent of the projection, Navier-Stokes or hydrostatic, which facilitates the comparison of the
two formalisms.

The basis $D^{\square}_4 = (\tilde{\mathbf{e}}_1, \tilde{\mathbf{e}}_2, \mathbf{e}_3, \mathbf{e}^{\square}_4)$, is dimensionally homogeneous and we have: $\tilde{\mathbb{L}}^{\square}\tilde{\mathbf{e}}_1 = -\tilde{\gamma}^{\square}\omega\tilde{\mathbf{e}}_2$ and $\tilde{\mathbb{L}}^{\square}\tilde{\mathbf{e}}_2 = (\tilde{\gamma}^{\square})^{-1}\omega\tilde{\mathbf{e}}_1$, where $\tilde{\gamma} = 1$ in the Navier-Stokes formalism. The third vector represents the thermal-wind balance, a generalization of geostrophy. Its eigen value vanishes with viscosity, it converges to a stationary state. We see that the first three vectors represent divergenceless motion as $(\tilde{\mathbf{e}}_i)^t \cdot (\mathbf{k}^t, 0) = 0, \forall i = 1, 2, 3$. They are identical in both formalisms. The fourth vector has a divergence:
$(\tilde{\mathbf{e}}_4)^t \cdot (\mathbf{k}^t, 0) = \tilde{\nu}(\tilde{\nu}^2 + \alpha)$ and $(\tilde{\mathbf{e}}^H_4)^t \cdot (\mathbf{k}^t, 0) = k_z$, in both formalisms and does therefore not participate in the dynamics. The error between the projections along the fourth direction, with a vanishing eigen value, is of first order $O(\gamma^1)$, if $f, N \ll \tilde{\nu}$ and of zeroth order $O(\gamma^0)$ if $\nu \to 0$ as the vectors $\mathbf{e}^H_4$ and $\mathbf{e}_4$ become orthogonal. A perturbation in the vertical acceleration triggers inertial-gravity waves in the Navier-Stokes formalism, whereas it has no effect in the hydrostatic case. If viscosity is zero the matrix $\tilde{\mathbb{L}}$ is not diagonalizable, has a Jordan-normal-form. I will therefore in the sequel discuss the case $\nu \to 0$ rather
than $\nu = 0$, which is also appropriate if numerical models are considered. Note that the first three components of $\tilde{\mathbf{e}}_1$ and $\mathbf{e}_3$ are aligned with $\mathbf{e}^P_1$ and those of $\tilde{\mathbf{e}}_2$ with $\mathbf{e}^P_2$. The matrix that transforms from the sub-basis $D = (\tilde{\mathbf{e}}_1, \tilde{\mathbf{e}}_2, \mathbf{e}_3)$ to the Cartesian coordinates is formed by the three basis vectors:

$$
\mathbb{E}_D = \begin{pmatrix} f\frac{k_y}{k_h} & \omega\frac{k_x}{k_h} & f\frac{k_y}{k_h} \\ -f\frac{k_x}{k_h} & \omega\frac{k_y}{k_h} & -f\frac{k_x}{k_h} \\ 0 & -\omega\frac{k_h}{k_z} & 0 \\ N\frac{k_h}{k_z} & 0 & -\frac{f^2}{N}\frac{k_z}{k_h} \end{pmatrix} \tag{37}
$$





For any vector in the subspace formed by $D$, the vertical acceleration to the horizontal counterpart is always at least of first
order $O(\gamma)$, as the first two components of $\tilde{\mathbf{e}}_2$ are orthogonal to the first two components of the two other basis vectors. A
result which can be also seen directly from the divergence free condition.

The validity of the hydrostatic approximation relies also on how strongly the modes $\tilde{\mathbf{e}}_1, \tilde{\mathbf{e}}_2$ on one side and $\mathbf{e}_3$ on the other,
get excited. For a comparison of the amplitudes of the forcing for each vector, the vectors are normalized by the square route
of twice the energy density, kinetic plus potential. The modules are:

$$\mathbb{e}_1 = \tilde{\gamma}^\square \mathbb{e}_2 = \sqrt{\frac{\alpha}{k_z^2}}, \qquad \mathbb{e}_3 = \gamma^{-1} \sqrt{\frac{f^2}{N^2}} \mathbb{e}_1 \tag{38}$$

and the energies of the first and the third mode are equal when the horizontal scale equals the Rossby radius of deformation.
Note that in the hydrostatic approximation the vertical velocity is not a dynamical variable and does not contribute to the kinetic
energy and, therefore, the energy density of $\tilde{\mathbf{e}}_2$ in the Navier-Stokes formalism agrees to the energy density of $\tilde{\gamma}^H \tilde{\mathbf{e}}_2$ in the
hydrostatic formalism. For small aspect ratios the energy of the balanced eigen-vector grows as $\gamma^{-1}$ with respect to the wave
modes.

A vector that represents an equilibrium, geostrophy in our case, is in the kernel of both operators, as its time evolution
vanishes. A conserved scalar quantity that is a linear combination of the components of the time evolution vector, divergence
and potential vorticity in our case, is represented by a vector that is in the kernel of the adjoint of the operators, as its dot
product with the time evolution vector vanishes. The adjoint operators are given by the transposed matrices $(\tilde{\mathbb{L}})^t$ and $(\tilde{\mathbb{L}}^H)^t$
for Navier-Stokes and the hydrostatic case, respectively. Note that neither of the operators is self-adjoint, that is, the matrices
are not symmetric. The kernel of the adjoint operators contains the wave vector $\mathbb{r}_3 = (\mathbf{k}, 0)^t$, which is therefore a conserved
quantity. This shows that the acceleration is divergence free in both formailsms, that the divergence of the velocity field is
conserved and the vanishing divergence is maintained during the evolution of the velocity field. When viscosity and diffusivity
vanish the vector $\mathbb{r}_4 = (k_y, -k_x, 0, -f k_z N^{-1})^t$ is also in the kernel of the adjoint hydrostatic-operator, this corresponds to
the conservation of linear potential-vorticity. Whereas, in the Navier-Stokes case we have $\tilde{\mathbb{L}}^t \mathbb{r}_4 = f \mathbb{r}_3$, this corresponds to the
conservation of linear potential-vorticity when the velocity field is divergence-free. As the velocity field is divergence free,
linear potential vorticity is conserved in both formalisms. It is interesting to note that for the Navier-Stokes formalism the
kernel is one dimensional as the sub-matrice corresponding to the zero eigen value has a Jordan normal form. The discussion
on the adjoint operators also shows that there are no more than two conserved quantities in the linear approximations.




## 6 Projection on the Basis of Eigen Vectors (S1)

The projection of a given acceleration $\tilde{\mathbf{a}}$ on the modes of the linear dynamics represents step S1, as defined in the introduction and shown in Fig. 2. The identities:

$$\tilde{\mathbf{e}}_1 - \mathbf{e}_3 = \frac{\alpha}{Nk_zk_h}\begin{pmatrix}0\\0\\0\\1\end{pmatrix}, \quad f^2k_z^2\tilde{\mathbf{e}}_1 + N^2k_h^2\mathbf{e}_3 = \frac{f\alpha}{k_h}\begin{pmatrix}k_y\\-k_x\\0\\0\end{pmatrix} \tag{39}$$

allow to write the extended projection operators that convert the forcing $\widehat{\mathbf{F}}$ to a divergenceless acceleration in the first three components and an unchanged buoyancy flux in the fourth: The projection operators given in Eqs. 25, 26 that project the generalized acceleration $\tilde{\mathbf{a}}$ written in terms of the basis $D^\square$ are:

$$_D\tilde{\mathbb{P}} = \frac{1}{k_h}\left[\frac{fk_z^2}{\alpha}\tilde{\mathbf{e}}_1 + \frac{N^2k_h^2}{f\alpha}\mathbf{e}_3\right](k_y,-k_x,0,0,0) + \frac{k_z^2}{\omega k^2 k_h}\tilde{\mathbf{e}}_2(k_x,k_y,-\frac{k_h^2}{k_z},0,0)$$
$$+k_zk_h\left[\frac{N}{\alpha}(\tilde{\mathbf{e}}_1-\mathbf{e}_3)-\frac{1}{\omega k^2}\tilde{\mathbf{e}}_2\right](0,0,0,1,0)$$
$$+\frac{Nk_zk_h}{\alpha}(\tilde{\mathbf{e}}_1-\mathbf{e}_3)(0,0,0,0,1). \tag{40}$$

$$_D\tilde{\mathbb{P}}^H = \frac{1}{k_h}\left[\frac{fk_z^2}{\alpha}\tilde{\mathbf{e}}_1 + \frac{N^2k_h^2}{f\alpha}\mathbf{e}_3\right](k_y,-k_x,0,0,0) + \frac{1}{\omega k_h}\tilde{\mathbf{e}}_2(k_x,k_y,0,0,0)$$
$$+k_zk_h\left[\frac{N}{\alpha}(\tilde{\mathbf{e}}_1-\mathbf{e}_3)-\frac{1}{\omega k_z^2}\tilde{\mathbf{e}}_2\right](0,0,0,1,0)$$
$$+\frac{Nk_zk_h}{\alpha}(\tilde{\mathbf{e}}_1-\mathbf{e}_3)(0,0,0,0,1) \tag{41}$$

In matrix form this is:

$$_D\tilde{\mathbb{P}} = \frac{1}{\alpha k_h}\begin{pmatrix}fk_z^2k_y & -fk_z^2k_x & 0 & Nk_zk_h^2 & Nk_zk_h^2\\ \frac{\alpha k_xk_z^2}{\omega k^2} & \frac{\alpha k_yk_z^2}{\omega k^2} & -\frac{\alpha k_h^2k_z}{\omega k^2} & -\frac{\alpha k_h^2k_z}{\omega k^2} & 0\\ \frac{N^2k_h^2k_y}{f} & -\frac{N^2k_h^2k_x}{f} & 0 & -Nk_zk_h^2 & -Nk_zk_h^2\end{pmatrix} \tag{42}$$

$$_D\tilde{\mathbb{P}}^H = \frac{1}{\alpha k_h}\begin{pmatrix}fk_z^2k_y & -fk_z^2k_x & 0 & Nk_zk_h^2 & Nk_zk_h^2\\ \frac{\alpha k_x}{\omega} & \frac{\alpha k_y}{\omega} & 0 & -\frac{\alpha k_h^2}{\omega k_z} & 0\\ \frac{N^2k_h^2k_y}{f} & -\frac{N^2k_h^2k_x}{f} & 0 & -Nk_zk_h^2 & -Nk_zk_h^2\end{pmatrix} \tag{43}$$

The operators only differ in the second line as all the differences are restricted to the second eigen-vector. The projection operators are the left pseudo-inverse of $\mathbb{E}_D$, that is $_D\tilde{\mathbb{P}}^\square\mathbb{E}_D = \mathbb{1}_D$, which means that the vector space spanned by the basis $D = (\tilde{\mathbf{e}}_1,\tilde{\mathbf{e}}_2,\mathbf{e}_3)$ is composed of divergenceless vectors.

### 6.1 Forcing of the Linear System

An exterior force $\widehat{\mathbf{F}} = (\widehat{F}_x,\widehat{F}_y,\widehat{F}_z,0,\widehat{B})^t$, was introduced in Section 5. The exterior forcing is projected on the basis $D$ by the projector $_D\tilde{\mathbb{P}}^\square$. I then determine the amplitude of the three basis vectors subject to different forcings. To this end, I write the



forcing vector as:

$$
\widehat{\mathbf{F}} = \begin{pmatrix} \frac{k_x \widehat{F}_d - k_y \widehat{F}_r}{k_h} \\ \frac{k_y \widehat{F}_d + k_x \widehat{F}_r}{k_h} \\ \widehat{F}_z \\ 0 \\ \widehat{B} \end{pmatrix},
\tag{44}
$$

where I split the horizontal forcing in a divergent and a rotational part, using the Helmholtz decomposition, to obtain:

$$
{}_D \tilde{\mathbb{P}} \widehat{\mathbf{F}} = -\widehat{F}_r \frac{1}{f\alpha} \begin{pmatrix} f^2 k_z^2 \\ 0 \\ N^2 k_h^2 \end{pmatrix} + \widehat{B} \frac{N k_h k_z}{\alpha} \begin{pmatrix} 1 \\ 0 \\ -1 \end{pmatrix} + \frac{k_z}{\omega k^2}(\widehat{F}_d k_z - \widehat{F}_z k_h)\begin{pmatrix} 0 \\ 1 \\ 0 \end{pmatrix},
\tag{45}
$$

$$
{}_D \tilde{\mathbb{P}}^H \widehat{\mathbf{F}} = -\widehat{F}_r \frac{1}{f\alpha} \begin{pmatrix} f^2 k_z^2 \\ 0 \\ N^2 k_h^2 \end{pmatrix} + \widehat{B} \frac{N k_h k_z}{\alpha} \begin{pmatrix} 1 \\ 0 \\ -1 \end{pmatrix} + \frac{\widehat{F}_d}{\omega}\begin{pmatrix} 0 \\ 1 \\ 0 \end{pmatrix} = {}_D \tilde{\mathbb{P}} \widehat{\mathbf{F}} + \frac{k_h}{\omega k^2}(\widehat{F}_d k_h + \widehat{F}_z k_z)\begin{pmatrix} 0 \\ 1 \\ 0 \end{pmatrix}.
\tag{46}
$$

The difference between the two operators is twofold: $\widehat{F}_z$ is ignored in the hydrostatic projection and for the projection of $\widehat{F}_d$ is of second order, $O(\gamma^2)$. Both projections are identical if rotational and buoyant forcings are considered. All the differences are only in the projections on the second eigen-vector. Therefore, the effect of all forcings on the balanced mode, which is the

third component in the above equations, is identical in both formalisms. That is, if forcings act differently in the hydrostatic approximation this difference does not directly affect the balanced mode. If the buoyancy frequency vanishes, $N \to 0$, all the energy from a rotational forcing goes to the first wave mode as no geostrophy is possible in this case. Furthermore, a divergent or vertical forcing does not act on the balanced mode. For a vertical forcing all the work is done on the wave part in the Navier-Stokes formalism, whereas it has no effect in the hydrostatic approximation. If $\widehat{F}_r f k_z = \widehat{B} N k_h$, the forcing is balanced and

all the work is done on the balanced mode.

## 6.2 Linearized Equation in the Eigen-Space

If the linear operator is restricted to a basis spanned by $D = (\tilde{\mathbf{e}}_1, \tilde{\mathbf{e}}_2, \mathbf{e}_3)$ it reads:

$$
{}_D \tilde{\mathbb{L}}^\square_D = \begin{pmatrix} -\tilde{\nu} & \omega & 0 \\ -(\tilde{\gamma}^\square)^2 \omega & -\tilde{\nu} & 0 \\ 0 & 0 & -\tilde{\nu} \end{pmatrix} \quad \text{and}
$$

$$
{}_D (\tilde{\mathbb{L}}^\square)^{-1}_D = \beta^\square \begin{pmatrix} -\tilde{\nu} & -\omega & 0 \\ (\tilde{\gamma}^\square)^2 \omega & -\tilde{\nu} & 0 \\ 0 & 0 & -(\tilde{\nu}\beta^\square)^{-1} \end{pmatrix}.
\tag{47}
$$

where $\beta^\square = (\tilde{\nu}^2 + (\tilde{\gamma}^\square \omega)^2)^{-1}$. The linearized dynamics consists in a damped thermal-wind balance along $\mathbf{e}_3$, for which the hydrostatic approximation is exact, and damped inertia-gravity waves around this balanced state along the directions $\tilde{\mathbf{e}}_1$ and $\tilde{\mathbf{e}}_2$.





The only difference between the Navier-Stokes and the hydrostatic approximation in the subspace spanned by $\{\tilde{\mathbf{e}}_1, \tilde{\mathbf{e}}_2\}$ lies in the frequencies, given in Eq. 34. Note that the correction $\tilde{\gamma}^\square$ appears only as a square and the approximations for the operator and the inverse operator are of second order: $O(\gamma^2)$.

## 7 Forced Motion (S2)

### 7.1 Slow Forced Motion, Ekman Dynamics

This section discusses the special case of the response of the linearized Navier-Stokes and hydrostatic operators subject to an almost time independent forcing. In geophysical fluid dynamics the large scale processes are evolving and interacting on a time-scale much slower than the wave frequencies $\tilde{\gamma}^\square \omega$. In this case the left-hand-side of Eq. 31 can be neglected. In the linear approximation an external (boundary) forcing can be balanced by a viscous force, the Coriolis force or both as in Ekman dynamics. Boundary conditions are local in physical space and therefore non-local in Fourier space, that is they are applied to different modes, which also interact through the boundary condition (see, e.g. Wirth (2005)). If the horizontal forcing is divergent, also the horizontal velocity field is and vertical velocities are induced, which represent Ekman pumping. If we suppose that the forcing $\widehat{\mathbf{F}}$ is weak, the linearized evolution Eq. 31 is suitable. Using the Boussinesq approximation, the forcing is represented as an acceleration. If the forcing is time independent, the stationary velocity field is restricted to the basis $D$ and can be obtained performing steps S1 & S2a through Eqs. 31, 42, 43 and 47:

$$\widehat{\mathbf{x}}_D^\square = -{}_D(\tilde{\mathbb{L}}^\square)_D^{-1}{}_D\tilde{\mathbb{P}}^\square \widehat{\mathbf{F}}. \tag{48}$$

The image of the operator ${}_D\tilde{\mathbb{P}}^\square$ is spanned by the basis $D$ for which ${}_D(\tilde{\mathbb{L}}^\square)_D^{-1}$ exists, if viscosity is non-vanishing.

### 7.1.1 Absolute Error

After having determined the amplitude of each mode due to the forcing (S1+S2a) the difference between the two formalisms in the basis $D$ is determined. The total error is:

$$E = \widehat{\mathbf{x}}_D^H - \widehat{\mathbf{x}}_D = \left[ -{}_D(\tilde{\mathbb{L}}^H)_D^{-1}{}_D\tilde{\mathbb{P}}^H + {}_D(\tilde{\mathbb{L}})_D^{-1}{}_D\tilde{\mathbb{P}} \right]\widehat{\mathbf{F}} = \left[ -{}_D(\tilde{\mathbb{L}}^H)_D^{-1} + {}_D(\tilde{\mathbb{L}})_D^{-1} \right]{}_D\tilde{\mathbb{P}}^H\widehat{\mathbf{F}} - {}_D(\tilde{\mathbb{L}})_D^{-1}\left[ {}_D\tilde{\mathbb{P}}^H - {}_D\tilde{\mathbb{P}} \right]\widehat{\mathbf{F}}$$

$$= \frac{\omega}{\alpha}\left[ (\beta - \beta^H)\begin{pmatrix} -\tilde{\nu}/\omega \\ 1 \\ 0 \end{pmatrix} - \gamma^2\beta^H\begin{pmatrix} 0 \\ 1 \\ 0 \end{pmatrix} \right]\left[ fk_z^2\widehat{F}_r + Nk_hk_z\widehat{B} \right] + \begin{pmatrix} 1 \\ \tilde{\nu}/\omega \\ 0 \end{pmatrix}\left[ (\beta^H - (\tilde{\gamma}^H)^{-2}\beta)\widehat{F}_d + \beta\frac{k_zk_h}{k^2}\widehat{F}_z \right] \tag{49}$$

We immediately see that the response to a stationary forcing of the balanced dynamics ($\mathbf{e}_3$) is identical in both formalisms. Furthermore, a balanced forcing with: $-fk_z\widehat{F}_r + Nk_h\widehat{B} = \widehat{F}_d = \widehat{F}_z = 0$, leads to a vanishing error. The last equality in the first line shows that the error can be separated in an evolution error, due to the differences in the wave frequencies (S2), and the error done by the hydrostatic projection (S1). The forcing $F_z$ is in the kernel of the hydrostatic projection (S1), only. This and the subsequent evolution (S2) of the projection error gives the error shown by the second part of the last term of the second line. The order can be written as $O(\gamma\widehat{F}_z + \gamma^2\widehat{F}_d + \gamma^2\widehat{F}_r + \tilde{\gamma}^3\widehat{B})$. If the vertical component of the forcing $\widehat{F}_z$ itself is of first order,





due to a boundary with a slope $\approx \gamma$, the approximation becomes second order, which shows that the hydrostatic approximation
has two ingredients: the horizontal scales are larger than the vertical and the vertical forcing is smaller than the horizontal.

For a vanishing viscosity $\tilde{\nu} \to 0$, the error associated with a divergent forcing $\widehat{F}_d$ is of second order $O(\tilde{\nu}^2)$, while it is of zero
order for the other forcings. Furthermore, a vertical forcing leads to a velocity of $\widehat{F}_z \frac{k_h k_z}{\omega^2 k^2} \tilde{\mathbf{e}}_1$ in the Navier-Stokes formalism,
while it has no effect in the hydrostatic approximation. As the vertical component of $\tilde{\mathbf{e}}_1$ is vanishing, so is the error in the
vertical component. For a strong viscous damping $\tilde{\nu} \to \infty$, present in numerical models just above the grid scale, where non-
hydrostatic effects are also important, we have $\beta = O(\tilde{\nu}^{-2})$ and the error is of order $O(\tilde{\nu}^{-1}\widehat{F}_z + \tilde{\nu}^{-1}\widehat{F}_d + \tilde{\nu}^{-2}\widehat{F}_r + \tilde{\nu}^{-2}\widehat{B})$.
For scales $f \approx \tilde{\nu}$ Ekman layer dynamics occurs.

### 7.1.2 Relative Error

We have seen above that the dynamics can be split into two parts: wave dynamics along the first two eigen-vectors and balanced
flow along the third eigen-vector. As the latter is free of error the fidelity of the hydrostatic approximation depends also on how
the energy is distributed among the eigen modes. This is achieved by considering the relative error,

$$E_R = \frac{|\widehat{\mathbf{E}}|}{|\widehat{\mathbf{x}}|}, \tag{50}$$

rather than the absolute error, with

$$\widehat{\mathbf{x}} = -_D\tilde{\mathbb{L}}_D^{-1}{}_D\tilde{\mathbb{P}}\widehat{\mathbf{F}} = \begin{pmatrix} -f^2 k_z^2 \tilde{\nu}\beta \\ f^2 k_z^2 \omega\beta \\ -N^2 k_h^2 \tilde{\nu}^{-1} \end{pmatrix} \frac{1}{f\alpha}\widehat{F}_r + \begin{pmatrix} -\tilde{\nu}\beta \\ \omega\beta \\ -\tilde{\nu}^{-1} \end{pmatrix} \frac{N k_h k_z}{\alpha}\widehat{B} + \begin{pmatrix} 1 \\ \tilde{\nu}/\omega \\ 0 \end{pmatrix} \frac{k_z \beta}{k^2}(\widehat{F}_d k_z - \widehat{F}_z k_h), \tag{51}$$

for the different types of forcing. Note that the absolute error is a vector while the relative error is a scalar. A constant forcing
on the wave mode leads to a finite amplitude motion, while in the balanced mode it is only countered by viscosity.

The balanced mode is excited by rotational and buoyancy forcing, only, and its amplitude compared to the amplitude of
the wave motion is proportional to $\tilde{\nu}^{-1}$, as it is damped by viscosity, only. Furthermore, Eq. 38 shows that the square root
of the energy of the third mode, as compared to the wave modes, grows with decreasing aspect ratio as $\gamma^{-1}$. The hydrostatic
approximation for the relative error is therefore $O(E_R) = \tilde{\nu}\gamma O(\widehat{\mathbf{E}})$, smaller, if the third component in Eq. 51 does not vanish,
that is if $N^2 k_h \widehat{F}_r \neq f k_z \widehat{B}$. This adds considerably to the validity of the hydrostatic approximation, for scales much larger than
the viscous cut-off. If the forcing is purely vertical, $\widehat{F}_z \neq 0$ and $\widehat{F}_r = \widehat{B} = \widehat{F}_d = 0$, the relative error is unity.

### 7.2 Projection-Evolution Operator (S2b)

After having discussed the case of a stationary force, I now consider the general case of a force varying in time. The linearized
equation governing the dynamics subject to a low amplitude forcing $\widehat{\mathbf{F}}(t)$ is given by Eq. 31. A forcing can only lead to
divergenceless acceleration and has therefore to be projected into the subspace of divergenceless functions by the corresponding
projector given in Eqs. 45 and 46. The dynamics takes place in the subspace spanned by $D$. Its solution, starting from the initial





condition $(\widehat{\mathbf{u}}(0), b(0))^t$, is:

$$
\begin{aligned}
\begin{pmatrix} \widehat{\mathbf{u}}^{\square}(t) \\ b^{\square}(t) \end{pmatrix} &= \mathbb{E}_D \exp({}_D\tilde{\mathbb{L}}_D^{\square}(t)) \left[ \int_0^t \exp({}_D\tilde{\mathbb{L}}_D^{\square}(-s)) \,{}_D\tilde{\mathbb{P}}^{\square}\widehat{\mathbf{F}}(s)ds + {}_D\tilde{\mathbb{P}}^{\square} \begin{pmatrix} \widehat{\mathbf{u}}(0) \\ b(0) \end{pmatrix} \right] \\
&= \mathbb{E}_D \int_0^t \exp({}_D\tilde{\mathbb{L}}_D^{\square}(t-s)) \,{}_D\tilde{\mathbb{P}}^{\square}\widehat{\mathbf{F}}(s)ds \\
&\quad + \mathbb{E}_D \exp({}_D\tilde{\mathbb{L}}_D^{\square}(t)) \,{}_D\tilde{\mathbb{P}}^{\square} \begin{pmatrix} \widehat{\mathbf{u}}(0) \\ b(0) \end{pmatrix},
\end{aligned}
\tag{52}
$$

with

$$
\exp({}_D\tilde{\mathbb{L}}_D^{\square}(t)) = e^{-\tilde{\nu}t} \begin{pmatrix} \cos(\tilde{\gamma}^{\square}\omega t) & (\tilde{\gamma}^{\square})^{-1}\sin(\tilde{\gamma}^{\square}\omega t) & 0 \\ -\tilde{\gamma}^{\square}\sin(\tilde{\gamma}^{\square}\omega t) & \cos(\tilde{\gamma}^{\square}\omega t) & 0 \\ 0 & 0 & 1 \end{pmatrix},
\tag{53}
$$

where I used: $\exp({}_D\tilde{\mathbb{L}}_D^{\square}(t))\exp({}_D\tilde{\mathbb{L}}_D^{\square}(-s)) = \exp({}_D\tilde{\mathbb{L}}_D^{\square}(t-s))$. Once the dynamics is projected on the basis $D$, the evolution is identical in both formalisms along $\mathbf{e}_3$ and the oscillations between $\tilde{\mathbf{e}}_1$ and $\tilde{\mathbf{e}}_2$ have a difference in the frequencies and the hydrostatic solution diverges as $\sin(\frac{\omega\gamma^2}{2}t)$, for a slowly varying forcing, if the forcing frequency $\zeta \ll \omega$. If the forcing has frequencies equal or close to $\tilde{\gamma}^{\square}\omega$, resonances can occur and the divergence can take place at a faster rate.

The unforced linear equations allow two types of motion: (decaying, if $\nu \neq 0$) internal waves with a frequency faster than $f$ and (decaying, if $\nu \neq 0$) thermal wind equilibrium. This dynamics is described by the second term on the r.h.s. of Eq. 52. The error done by the hydrostatic approximation in the basis $D$, starting from rest and subject to a time-dependent force $\widehat{\mathbf{F}}(t)$ is:

$$
\triangle\widehat{\mathbf{F}} = \int_0^t \left[ \exp({}_D\tilde{\mathbb{L}}_D^H(t-s)) \,{}_D\tilde{\mathbb{P}}^H - \exp({}_D\tilde{\mathbb{L}}_D(t-s)) \,{}_D\tilde{\mathbb{P}} \right] \widehat{\mathbf{F}}(s)ds
\tag{54}
$$

### 7.3 Periodic Forcing

When solving linear equations, solutions can be considered independently for each wave vector $\mathbf{k}$ and frequency $\zeta$. For a periodic forcing with frequency $\zeta$, the evolution Eq. 31 in the basis $D$ is:

$$
\partial_t \widehat{\mathbf{x}}^{\square} = \tilde{\mathbb{L}}_D^{\square}\widehat{\mathbf{x}}^{\square} + \widehat{\mathbf{F}}_D\cos(\zeta t).
\tag{55}
$$

The general solution is:

$$
\widehat{\mathbf{x}}^{\square} = \exp({}_D\tilde{\mathbb{L}}_D^{\square,\zeta}(t))\widehat{\mathbf{F}}_D = \int_0^t \exp({}_D\tilde{\mathbb{L}}_D^{\square}(t-s))\cos(\zeta s)ds\,\widehat{\mathbf{F}}_D.
\tag{56}
$$

Before discussing the general case I examine the case of a very small viscosity ($\tilde{\nu}^2 \ll (\tilde{\gamma}^{\square}\omega)^2, \zeta^2$). The solution is:

$$
\widehat{\mathbf{x}} = \frac{1}{\zeta^2 - (\tilde{\gamma}^{\square}\omega)^2} \begin{pmatrix} \zeta\sin(\zeta t) - a\tilde{\gamma}^{\square}\omega\sin(\tilde{\gamma}^{\square}\omega t) & \omega(-\cos(\zeta t) + a\cos(\tilde{\gamma}^{\square}\omega t)) & 0 \\ (\tilde{\gamma}^{\square})^2\omega(\cos(\zeta t) - a\cos(\tilde{\gamma}^{\square}\omega t)) & \zeta\sin(\zeta t) - a\tilde{\gamma}^{\square}\omega\sin(\tilde{\gamma}^{\square}\omega t) & 0 \\ 0 & 0 & \frac{\zeta^2 - (\tilde{\gamma}^{\square}\omega)^2}{\zeta}\sin(\zeta t) \end{pmatrix} \widehat{\mathbf{F}}_D.
\tag{57}
$$





The constant $a$ is there to satisfy the initial condition. For high-frequency forcing $\zeta^2 \gg (\tilde{\gamma}^\square\omega)^2$ there are no resonances and the part independent of the initial condition, which is "forgotten" after a long time $t \gg \tilde{\nu}^{-1}$, is:

$$\widehat{\mathbf{x}}^\square \quad = \quad \exp(_D\tilde{\mathbb{L}}_D^{\square,\zeta}(t))\widehat{\mathbf{F}}_D = \zeta\sin(\zeta t)\widehat{\mathbf{F}}_D. \tag{58}$$

In this case the operator $\tilde{\mathbb{L}}_D^\square$ has no time to act on the dynamics.

The general solution for $t \gg \tilde{\nu}^{-1}$, when the initial condition is forgotten, is:

$$\widehat{\mathbf{x}}^\square = \begin{pmatrix} \zeta\tilde{s}^\square & -\omega\tilde{c}^\square & 0 \\ (\tilde{\gamma}^\square)^2\omega\tilde{c}^\square & \zeta\tilde{s}^\square & 0 \\ 0 & 0 & \tilde{b} \end{pmatrix}\mathbf{F}_D \tag{59}$$

with:

$$\tilde{s}^\square \quad = \quad \tilde{q}^\square\left[\frac{\tilde{\nu}}{\zeta}(\tilde{\nu}^2 + \zeta^2 + (\tilde{\gamma}^\square\omega)^2)\cos(\zeta t) + (\tilde{\nu}^2 + \zeta^2 - (\tilde{\gamma}^\square\omega)^2)\sin(\zeta t)\right] \tag{60}$$

$$\tilde{c}^\square \quad = \quad \tilde{q}^\square\left[-(\tilde{\nu}^2 - \zeta^2 + (\tilde{\gamma}^\square\omega)^2)\cos(\zeta t) - 2\tilde{\nu}\zeta\sin(\zeta t)\right] \tag{61}$$

$$(\tilde{q}^\square)^{-1} \quad = \quad (\tilde{\nu}^2 + (\zeta + \tilde{\gamma}^\square\omega)^2)(\tilde{\nu}^2 + (\zeta - \tilde{\gamma}^\square\omega)^2)$$

$$= \quad \tilde{\nu}^4 + 2\tilde{\nu}^2(\zeta^2 + (\tilde{\gamma}^\square\omega)^2) + (\zeta^2 - (\tilde{\gamma}^\square\omega)^2)^2 \tag{62}$$

$$\tilde{b} \quad = \quad \frac{\tilde{\nu}\cos(\zeta t) + \zeta\sin(\zeta t)}{\tilde{\nu}^2 + \zeta^2}. \tag{63}$$

For a stationary forcing, $\zeta = 0$, the results from Subsection 7.1 are obtained and if $\tilde{\nu}^2 \ll (\tilde{\gamma}^\square\omega)^2, \zeta^2$, the solution given in Eq. 58 is recovered. The wave frequency only appears as a square, which assures that the error of the projection-evolution operator is of second order. The eigen values of the evolution matrix are:

$$\lambda_{W1/W2}^\square \quad = \quad \zeta\tilde{s}^\square \pm i\tilde{\gamma}^\square\omega\tilde{c}^\square \qquad \text{and} \qquad \lambda_b = \tilde{b}. \tag{64}$$

The corresponding eigen vectors are $\mathbf{e}_{1,2}^\square$ and $\mathbf{e}_3$. Resonances are efficiently damped by viscosity if for a forcing at the resonant frequency, $\zeta^2 = \omega^2$, the viscous time-scale is larger than the resonance period, $\tilde{\nu}^2 > \omega^2$.

### 7.3.1 Absolute Error

The difference between the two evolution operators in the case of a long viscous damping time $\tilde{\nu}^2 \ll \zeta^2, (\tilde{\gamma}^\square\omega)^2$ is:

$$\mathbb{G}_\zeta^0 = \frac{\gamma^2}{(\zeta^2 - (\tilde{\gamma}^\square)^2\omega^2)(\zeta^2 - \omega^2)}\begin{pmatrix} \zeta\omega^2\sin(\zeta t) & -\omega^3\cos(\zeta t) & 0 \\ \zeta^2\omega\cos(\zeta t) & \zeta\omega^2\sin(\zeta t) & 0 \\ 0 & 0 & 0 \end{pmatrix}. \tag{65}$$

The hydrostatic approximation is of second order. In the general case the difference between the two evolution operators is:

$$\mathbb{G}_\zeta^\nu = \begin{pmatrix} \zeta(\tilde{s}^H - \tilde{s}) & -\omega(\tilde{c}^H - \tilde{c}) & 0 \\ \tilde{\gamma}^2\omega\tilde{c}^H - \omega\tilde{c} & \zeta(\tilde{s}^H - \tilde{s}) & 0 \\ 0 & 0 & 0 \end{pmatrix}. \tag{66}$$





All the components of the matrix are at least of second order, $O(\gamma^2)$.

The error done by the hydrostatic approximation with a periodic forcing of frequency $\zeta$ is:

$$\widehat{\mathbf{E}}^\nu_\zeta = \left[ \mathbb{G}^\nu_\zeta\,_D\widetilde{\mathbb{P}}^H + \exp(_D\widetilde{\mathbb{L}}^\zeta_D(t))(_D\widetilde{\mathbb{P}}^H - _D\widetilde{\mathbb{P}}) \right] \widehat{\mathbf{F}}. \tag{67}$$

It has two terms: the first is due to the different evolution in the two formalisms (see Eq. 66) and the second due to the difference in the projections (see Eq. 46). The explicit form is:

$$\widehat{\mathbf{E}}^\nu_\zeta = \frac{k_z}{\alpha}(-f k_z \widehat{F}_r + k_h \widehat{B}) \begin{pmatrix} \zeta(\tilde{s}^H - \tilde{s}) \\ (\tilde{\gamma}^\square)^2 \omega \tilde{c}^H - \omega \tilde{c} \\ 0 \end{pmatrix} + \widehat{F}_d \left[ \begin{pmatrix} -(\tilde{c}^H - \tilde{c}) \\ \frac{\varsigma}{\omega}(\tilde{s}^H - \tilde{s}) \\ 0 \end{pmatrix} + \frac{k_h^2}{k^2} \begin{pmatrix} -\tilde{c} \\ \frac{\varsigma}{\omega}\tilde{s} \\ 0 \end{pmatrix} \right] - \widehat{F}_z \frac{k_h k_z}{k^2} \begin{pmatrix} -\tilde{c} \\ \frac{\varsigma}{\omega}\tilde{s} \\ 0 \end{pmatrix}. \tag{68}$$

For the case of a long viscous damping time I obtain:

$$\widehat{\mathbf{E}}^0_\zeta = \frac{\gamma^2 \omega \zeta}{(\zeta^2 - (\omega^H)^2)(\zeta^2 - \omega^2)} \left[ \frac{k_z}{\alpha}(-f k_z \widehat{F}_r + k_h \widehat{B}) \right] \begin{pmatrix} \omega \sin(\zeta t) \\ \zeta \cos(\zeta t) \\ 0 \end{pmatrix}$$

$$+ \frac{1}{\zeta^2 - \omega^2} \left[ \widehat{F}_d \left( \frac{\gamma^2 \omega}{\zeta^2 - (\omega^H)^2} + \frac{k_h^2}{\omega k^2} \right) - \widehat{F}_z \frac{k_h k_z}{\omega k^2} \right] \begin{pmatrix} -\omega \cos(\zeta t) \\ \zeta \sin(\zeta t) \\ 0 \end{pmatrix}. \tag{69}$$

This shows, once more, that the third component, the balanced dynamics, is free of error. The scaling of the error with the aspect ratio is:

$$O(\widehat{\mathbf{E}}) = O(\gamma^2 \widehat{F}_r + \gamma^3 \widehat{B} + \gamma^2 \widehat{F}_d + \gamma \widehat{F}_z). \tag{70}$$

If the forcing evolves much slower than the buoyancy frequency the dynamics is close to the stationary state calculate in Subsection 7.1. If the forcing occurs at the wave frequency of either formalism, resonances or near resonances dominate the linear dynamics. The hydrostatic approximation breaks down as $\omega^H \neq \omega$ if $k_h \neq 0$ and a resonance occurs in one formalism but not in the other. During resonances or close resonances the scaling is still given by Eq. 70 but the prefactor diverges. Note that if $\tilde{\nu}^2 \gg (\tilde{\gamma}^\square \omega)^2$ resonances are efficiently damped.

### 7.3.2 Relative Error

To obtain the relative error the procedure is identical to the case of stationary forcing considered in subsection 7.1.2. We compare the absolute error calculated above to the solution of the Navier-Stokes model, using Eq. 59,

$$\widehat{\mathbf{x}}^N = \widehat{F}_r \frac{1}{f\alpha} \begin{pmatrix} -f^2 k_z^2 \zeta \tilde{s}^N \\ -f^2 k_z^2 \omega^2 \tilde{c}^N \\ N^2 k_h^2 \tilde{b} \end{pmatrix} + \widehat{B} \frac{k_h k_z}{\alpha} \begin{pmatrix} \zeta \tilde{s}^N \\ \omega^2 \tilde{c}^N \\ -\tilde{b} \end{pmatrix}$$

$$+ \left( \widehat{F}_d \frac{k_z^2}{k^2} - \widehat{F}_z \frac{k_z k_h}{k^2} \right) \begin{pmatrix} -\tilde{c}^N \\ \zeta \tilde{s}^N \\ 0 \end{pmatrix} \tag{71}$$





which is taken as the reference. Calculations confirm the results from subsection 7.1.2 also for the case of periodic forcing, except that prefactors in the scaling can diverge due to resonances.

## 7.4   Non-linear Terms

Formally the non-linear terms can be added to the forcing in Eq. 52 by $\widehat{\mathbf{F}} \to \widehat{\mathbf{F}} + \widehat{\mathbf{F}}_{\mathrm{nl}}$. Using Eqs. 13 to 16 I obtain:

$$\widehat{\mathbf{F}}_{\mathrm{nl}} = -i \begin{pmatrix} k_x \widehat{u_x^2} + k_y \widehat{u_x u_y} + k_z \widehat{u_x u_z} \\ k_x \widehat{u_x u_y} + k_y \widehat{u_y^2} + k_z \widehat{u_y u_z} \\ k_x \widehat{u_x u_z} + k_y \widehat{u_y u_z} + k_z \widehat{u_z^2} \\ k_x \widehat{u_x b} + k_y \widehat{u_y b} + k_z \widehat{u_z b} \end{pmatrix} \tag{72}$$

Due to the zero divergence of the velocity field, the horizontal velocity components are at least a factor $\gamma$ larger than the vertical velocity, $\widehat{u_x} \sim \widehat{u_y} \sim \gamma^{-1} \widehat{u_z}$. If this property is conserved by the non-linear term, the hydrostatic approximation is of second order $O(\gamma^2)$, following Eq. 70.

## 8   Discussion

The hydrostatic approximation is exact on the balanced mode and if this mode receives all the energy by a given forcing then
the approximation is exact. If the forcing strongly excites wave-motion then the deficiencies of the hydrostatic approximation in predicting the wave dynamics determines its overall quality. The hydrostatic approximation fails if the forcing is a vertical acceleration which is not a dynamical variable in the hydrostatic approximation and is therefore ignored, while in the Navier-Stokes formalism it generates horizontal velocities of first order in the scale ratio, $O(\gamma)$.

      In my evaluation, done using the techniques T1, T2 and T3 in Fourier space, the calculation is done separately for every three
dimensional wave vector $\mathbf{k}$. The projection step, S1, is instantaneous, it does not depend on the temporal variation of the forcing. Whereas the evolution step, S2, asks for integrating the evolution equations in time. In the linear approximation forcings at different frequencies superpose without interacting and step S2 can be calculated for each forcing frequency separately. The quality of the hydrostatic approximation depends on the Coriolis parameter $f$, the buoyancy frequency $N$ and the viscosity $\nu$. It has to be determined for every three dimensional wave vector $\mathbf{k}$ and forcing vector $\widehat{\mathbf{F}}$ and frequency $\zeta$ (see Fig 2).

The mathematical formalism to evaluate the hydrostatic approximation was developed and applied in the previous sections. It is now important to compare the deficiencies of the hydrostatic approximation to other uncertainties, to evaluate its fidelity. The previous discussion shows that viscosity and damping can mask the differences in the wave frequencies between the two formalisms. This will be assessed in Subsection 8.1. The wave packets move at the group velocity and are advected by the mean flow, which shows internal variability. In Subsection 8.2 I compare this Doppler effect to the error due to the hydrostatic
approximation. Every measurement or observation is limited in time and space. Subsection 8.3 applies the Heisenberg-Gabor uncertainty principle to discuss for which processes the difference between the two formalisms is actually detectable, given the finite space-time observations.





## 8.1 Masking of Resonance Differences by Friction

I first note that the amplitude of the balanced mode is bounded by $(\tilde{\gamma}^{\Box}\omega)^{-1}$. Resonances occur when $\sqrt{(\zeta^2 - (\tilde{\gamma}^{\Box}\omega)^2)^2} \ll$
$(\tilde{\gamma}^{\Box}\omega)^2$. For not too small viscous damping, $(\zeta^2 - (\tilde{\gamma}^{\Box}\omega)^2)^2 \gg 2(\tilde{\gamma}^{\Box}\omega)^2\tilde{\nu}^2$, Eqs. 60 and 61 approach:

$$\tilde{s}^{\Box} \approx \frac{1}{2\tilde{\gamma}^{\Box}\omega\tilde{\nu}}\cos(\tilde{\gamma}^{\Box}\omega t), \tag{73}$$

$$\tilde{c}^{\Box} \approx \frac{-1}{2\tilde{\gamma}^{\Box}\omega\tilde{\nu}}\sin(\tilde{\gamma}^{\Box}\omega t). \tag{74}$$

The amplitude of the wave motion is bounded by viscosity and diverges as $\tilde{\nu}^{-1}$. There is no significant difference between the two formalisms. For smaller viscosities $(\zeta^2 - (\tilde{\gamma}^{\Box}\omega)^2)^2 \ll 2(\tilde{\gamma}^{\Box}\omega)^2\tilde{\nu}^2$:

$$\tilde{s}^{\Box} \approx \frac{2\tilde{\gamma}^{\Box}\omega\tilde{\nu}}{(\zeta^2 - (\tilde{\gamma}^{\Box}\omega)^2)^2}\cos(\tilde{\gamma}^{\Box}\omega t), \tag{75}$$

$$\tilde{c}^{\Box} \approx \frac{2\tilde{\gamma}^{\Box}\omega\tilde{\nu}}{(\zeta^2 - (\tilde{\gamma}^{\Box}\omega)^2)^2}\sin(\tilde{\gamma}^{\Box}\omega t). \tag{76}$$

The amplitude of the wave motion shows a quadratic divergence: $O\left(\zeta \pm (\tilde{\gamma}^{\Box}\omega)\right)^{-2}$ and resonances can occur in one formalism and not in the other when:

$$\gamma^2 \ll \frac{\tilde{\nu}^2}{\omega^2}. \tag{77}$$

which occurs at scales: $k \gg k_r = (\gamma\omega/\nu)^{1/2}$. If a hyper- or hypo dissipation operators $\nu_\alpha\nabla^{2\alpha}$ are used (hyper-viscosity, $\alpha = 1$ corresponding to Laplacian dissipation and $\alpha = 0$ to scale independent bottom friction), this occurs at scales $k \gg k_r = (\gamma\omega/\nu_\alpha)^{1/(2\alpha)}$.

In simulations of the ocean dynamics we either resolve the turbulent mixed layer dynamics at the surface and the bottom, which asks for a resolution on the meter scale in the vertical and the horizontal, or we parameterize it, which asks for vertical
viscosity above $\nu_z > 10^{-2}$ m$^2$s$^{-1}$. In numerical models of the ocean dynamics resolution and viscous damping are very different in the horizontal and the vertical. If the model is fine resolution, an isotropic eddy viscosity is chosen based on the Smagorinsky large-eddy simulation model. Its viscous damping time is $\tilde{\nu} = C_s^2 u/\Delta x$. A Smagorinsky constant $C_s = 0.16$, a current speed of $10^{-1}$ m s$^{-1}$ and a grid scale $\Delta x$, leads to $\tilde{\nu} \approx 3 \cdot 10^{-3}$ m s$^{-1}/(\Delta x)$. If we suppose that $\omega > 10^{-4}$ s$^{-1}$ Eq. 77 gives: $\gamma \ll \frac{30\text{m}}{\Delta x}$, meaning that the difference between the resonant frequencies in both formalisms is hidden by viscosity
if the vertical scale is below 30 m, which is a typical scale for the oceanic Ekman-layer. This shows that viscosity masks differences between the two formalisms in the bottom and surface Ekman-layer dynamics. At larger horizontal scales friction is parameterized through turbulent bottom friction. A bottom friction coefficient of $\approx 10^{-3}$, a current speed of $10^{-1}$ m s$^{-1}$ and a resonance frequency $\omega \approx 10^{-4}$ s$^{-1}$ shows that bottom friction is sufficient to hide the difference in resonance frequencies only if the ocean layer thickness in meters is below $\gamma^{-1}$. The above shows that neither viscous nor bottom friction can mask
the difference in the resonance frequencies for large scale dynamical processes.





## 8.2 Comparison to Doppler Shift and effective Coriolis parameter

Horizontal waves evolve in a medium that is moving with a velocity, $u$, which leads to a Doppler shift in the frequency at an Eulerian point. If the velocity, $u$, has a turbulent variability $\Delta u$, due to the turbulent dynamics of the flow field, also the Doppler shift in the frequency is modified and the modification can be compared to the error done by the hydrostatic approximation. The difference in the horizontal wave velocity, $c$, between the two formalisms is $\Delta c = \gamma c = \gamma \omega/k_h = \omega/k_z$, which equals the vertical phase speed. The difference between the formalisms is significant if $\Delta c > \Delta u$, that is if $\gamma > \Delta u/c$. In the Strait of Gibraltar the uncertainty in the horizontal advection due to natural variability, is on the order of $\Delta u = 0.1$ m s$^{-1}$ and $c = 1$ ms$^{-1}$, so that for $\gamma < 0.1$ the hydrostatic approximation can not be distinguished from the Navier-Stokes solution. With a depth of around 300 m the horizontal wave length has to be below 3000 m which asks for a resolution of 300 m or finer, to distinguish between the two formalisms. The same can be applied to the group velocity of a moving wave packet, but as it is lower than the phase speed it is even more affected by the Doppler shift. We see that internal variability of the horizontal velocity masks efficiently the differences between the two formalisms.

Gradients of the horizontal velocities lead to vorticity, $\zeta$, and change the effective Coriolis parameter $f_{\text{eff}} = \sqrt{1 + \zeta/f}\,f$ (for $f > 0$) (see Wirth (2013)). If the Rossby number $\zeta/f > 2\gamma^2\omega^2/f^2$, the change in the effective Coriolis parameter is larger than the difference in the formalisms. For oceanic values of $N^2/f^2 = 100$ and $\gamma < 0.01$ changes in the formalisms for Rossby numbers larger than $2 \cdot 10^{-2}$ are masked. A horizontal velocity difference across the strait of Gibraltar of 0.3 m s$^{-1}$, leading to a Rossby number of 0.1, masks the changes in the frequencies of the formalisms for a horizontal wave-length above 3 km.

## 8.3 Uncertainty estimation

This section discusses the estimation of uncertainty in time and space based on the theory of the Heisenberg-Gabor limit (see Folland and Sitaram (1997) for a general discussion and Mordant (2010) for an application similar to the present). If considering the time-frequency formalism it shows, that the resolution of the frequency (described by the standard deviation, $\sigma_f$, in the frequency domain of the normalized signal) and the length of the signal (described by the standard deviation of the temporal extent of the signal $\sigma_t$) are restraint by:

$$\sigma_f \sigma_t \geq \frac{1}{4\pi}, \tag{78}$$

equality being obtained only by a Gaussian function. Note, that this applies to perfect data not disturbed by superposed signals or noise, so it is independent of the quality of the observation or the model, it is purely theoretical.

The difference in frequency between the two formalisms is $\sigma_f = \omega\gamma/2$. To discriminate between the two frequencies in a signal, the Heisenberg-Gabor limit requires an observation time $\sigma_t \geq (2\pi\omega\gamma)^{-1}$. For short waves as for example wave-trains generated by tidal forcing in the Strait of Gibraltar we have $\omega \approx 10f$. A resolved traveling wave asks for a grid scale which is at least one tenths of the wave length and if the horizontal resolution is about 10 times the water depth we obtain $\gamma > 0.01$. This leads to an observation time of the phenomenon longer than the Coriolis period, which exceeds the lifetime of waves generated by tidal motion in, for example, the Strait of Gibraltar. This means that even in the absence of noise, currents and other phenomena it is mathematically impossible to distinguish between the hydrostatic and the Navier-Stokes



wave frequency. Furthermore, during this time, the advective and the group velocity will have transported the wave-packet several tens of kilometers to regions where the depth and stratification largely differ, further altering the wave frequency and the phase and group velocity.

If waves with larger horizontal extensions are considered $\gamma \gg 0.01$ with $\omega \approx f$ the observation time to distinguish between the hydrostatic and the Navier-Stokes wave-frequencies increases to several weeks. This means the differences between the two formalisms can not be detected in the temporal domain.

If the spatial resolution is fine enough so that vertical convection can be explicitly resolved $\Delta x = \Delta z < 10$ m we have $\gamma \approx 1$, the domain is essentially unstratified, so that $\omega \approx f$ and we obtain $t_s \geq \sigma_t = \frac{1}{2\pi f} = 2$ days, which is about the time an oceanic deep convection plume takes to go from surface to the bottom. So also in the case of oceanic deep convection the difference between the two projection-evolution operators can not be detected. The difference in the projection operator in the two formalisms might be significant due to the advective transport of vertical velocity, which is present in the Navier-Stokes formalism but neglected in the hydrostatic approximation.

In the spatial domain the wave lengths can be considered in the vertical and the horizontal domain. To calculate the difference in wave vectors $\mathbf{k}^H$ and $\mathbf{k}$ between both formalisms for a given wave-frequency, the equation: $\omega^H(\mathbf{k}^H) = \omega(\mathbf{k})$ has to be solved. This is done separately in the horizontal (supposing $k_z^H = k_z$) and the vertical (supposing $k_h^H = k_h$). In the horizontal the relation between the hydrostatic and the Navier-Stokes horizontal wave-number is:

$$(k_h^H)^2 - (1 - \frac{f^2}{N^2})\frac{k_z^2}{k^2}k_h^2 = 0. \tag{79}$$

For a strong stratification and small $\gamma$ this approaches $(k_h^H)^2 = (1 - \gamma^2)k_h^2$. Using the Heisenberg-Gabor limit means that the observation length has to exceed $\gamma^{-2}/(4\pi)$ the wave length, which means that for a rather large $\gamma \approx 0.1$, we need at least 10 wavelength to distinguish between the two formalisms and $\gamma \approx 0.01$ asks for an observation of around 1000 wave periods, no wave packets of such extension are observed in the ocean.

In the vertical the solution is

$$(1 - \frac{f^2}{N^2})(k_z^H)^2 - (1 + \gamma^2)k_z^2 = 0. \tag{80}$$

For the leading vertical modes there are only from one to a few half periods imposed by the depth of the domain. The question I answer is by how much does the depth has to vary, to perfectly compensate the differences of the two formalisms? If $N = 10f$ a typical value for the ocean with a buoyancy frequency of a few hours and $\gamma = 0.01$ the difference is 1 %. With a vertical extension of 500 m of the phenomena, as for example in the strait of Gibraltar, this asks for a vertical resolution of the topography below 5 m. Such precision is not possible with today's models, as, for example, the topography or the thermocline depth largely exceed 5 m from one horizontal grid-point to the next. So also in this case the difference in the projection-evolution operator between the two formalisms is insignificant.

Finally I consider by how much the buoyancy frequency has to differ to compensate the error done by the hydrostatic approximation. To make the wave frequency of both formalisms agree at the same wave vector, the difference in the buoyancy frequency is $(N^H)^2 = (N^2 - f^2)/(1 + \gamma^2)$, for $N^2 = 100f^2$ and $\gamma = 0.1$ this leads to a difference in buoyancy frequency of less than 2 %, far below the precision of today's models or observations.



The above discussions on the uncertainty of the horizontal, vertical wave-number and the buoyancy frequency demonstrate that the difference in the projection-evolution operators of the formalisms is not significant in realistic simulations of ocean dynamics, for small amplitude, linearized, motion. Only non-linear or externally forced processes can lead to configurations that project differently into the subspace of divergenceless fields, in the two formalisms.

## 9 Conclusions

The present paper displays a theoretical framework to investigate the differences between the Navier-Stokes model and the hydrostatic approximation. To investigate the fidelity of the hydrostatic approximation I constructed a four dimensional local projection of the dynamics in Fourier space, by separating the buoyant and dynamic vertical acceleration (T1+T2) and combined it to the linear projection-evolution operator (T3). The linearized dynamics in the hydrostatic and Navier-Stokes formalism is compared. It shows, that the hydrostatic approximation relies on two properties, the difference in the eigen vectors of the linear operator between the two formalisms and the difference in the corresponding eigen values. The former shows in the projection on the eigen vectors (S1) while the latter in the evolution in the eigen spaces (S2). The projection depends on the Coriolis frequency $f$, the buoyancy frequency $N$ and the three dimensional wave vector $\mathbf{k}$. The projection-evolution operator depends on the same parameters, plus the forcing frequency $\zeta$.

The eigen spaces of the wave dynamics (2D) and the balanced dynamics (1D) are identical between both formalisms. The eigen value for the balanced flow agrees between the two formalisms whereas the eigen values in the wave dynamics differ due to the different wave-frequencies. Therefore: for the projection-evolution operator (S2), the hydrostatic approximation is exact if the balance dynamics is considered, whereas it differs to second order $O(\gamma^2)$ for the wave dynamics. The fourth eigen vector is in the kernel of the corresponding projection. It is different in both formalisms to zeroth order $O(\gamma^0)$, but as the corresponding eigenvalue is vanishing, it does not participate in the dynamics.

The difference in the projection (S1) between the two formalisms is always along the second vector of the eigen space of the wave dynamics ($\tilde{\mathbf{e}}_2$). It corresponds to horizontally divergent flow, with a vanishing vertical component of the vorticity vector and is caused either by a purely vertical force, with error $O(\gamma)$, or a horizontally divergent force, with error $O(\gamma^2)$. The projection of a rotational forcing and a buoyant forcing on the eigen vectors is identical in both formalisms. This makes the hydrostatic approximation to be first order, except if the vertical dynamical-acceleration is of first order, which would make the hydrostatic approximation to be second order $O(\gamma^2)$. An expansion of the non-linear terms in powers of $\gamma$ suggests that its vertical acceleration is indeed first order.

The error in the projection-evolution operator is second order $O(\gamma^2)$ for the wave dynamics and exact for the balanced flow. The validity of the hydrostatic approximation relies not only on the order of approximation in the scale-ratio, $\gamma$, but also on the amount of energy that is found in the subspaces spanned by the wave modes and the balanced mode. As the dynamics of the latter is perfectly described by the hydrostatic approximation. For fast motion, if the frequency of the dynamics is larger than the minimum of either $f$ and $N$, resonances can occur and the hydrostatic approximation breaks down even if the projection-evolution operator is formally second order. But for slower and slower motion, of frequency $\zeta$, relatively more and more energy





goes into the balanced mode, for which the hydrostatic approximation is exact. I showed that this depends also on the viscous damping. Another important result is that I manage to write the linear projection-evolution operators so that the lowest power of the wave frequency $\omega$ is quadratic, which assures that the approximation for the linear projection-evolution operator is second order.

There are two important points one has to keep in mind: first the hydrostatic approximation is a singular perturbation of the full Navier-Stokes equations, as the highest (temporal) derivative in the equation for the vertical velocity is omitted and expansions in small parameters are no guarantee for the smallness of the error (see i.e. Bender et al. (1999)). Second, the hydrostatic approximation breaks down where neither: rotation nor stratification nor scale-ratio constrain the fluid, that is, in fast moving unstratified fluid dynamics. Where, when and how such areas are spontaneously created by the turbulent forced dynamics or external forcing, depends on the situation considered. For these reasons dedicated numerical experiments of an oceanic phenomena have to be employed to determine the fidelity of the hydrostatic approximation for the scales and phenomena studied.

Considering a non-vanishing viscosity is of importance as in numerical models the non-hydrostatic effects are present at the smallest scales of the model at which the numerical viscosity parameter has a dominant effect. It is straightforward to extend the calculations to an anisotropic viscosity, which vertical values much smaller than the horizontal counter-parts, usually employed due to the disparate resolution in the vertical and horizontal. Such anisotropy, however, hinders the fully three-dimensional dynamics to commence. Comparing the impact of anisotropic friction to the hydrostic approximation has not been done here, as in fully non-hydrostatic simulations, that resolve the dynamics down to the meter scale, an anisotropic viscous damping lags justification.

It is shown that if the vertical turbulent-viscosity parameterizes the turbulent Ekman-layer dynamics the viscosity is high enough to mask the differences between the two formalisms in the wave frequencies at scales below the Ekman layer thickness ($\approx 30$ m). If it is lower $\nu_z < 10^{-2}\mathrm{m}^2\mathrm{s}^{-1}$ the vertical turbulent-exchange of momentum has to be resolved explicitly, which asks for a grid scale in the vertical and horizontal which is at one meter or finer. At these scales the free surface variations become important leading to Langmuir cells (see McWilliams et al. (1997)). For an explicit representation of these processes the fully three dimensional Navier-Stokes equations with a free surface should be used. Such models will not be available for basin wide simulations in the near future and parameterizations for surface processes have to be used.

At larger scales the Doppler shift, the change in the effective Coriolis parameter due to unresolved processes and the internal variability of the dynamics, and also the variation of the topography do not allow to discriminate between the two formalisms in realistic ocean simulations. Furthermore, the uncertainty obtained through the Heisenberg-Gabor limit shows that the observation time of propagating inertia-gravity waves does not allow to discriminate between the two formalisms. In conclusion: if the vertical viscosity is larger than $\nu_z > 10^{-2}\mathrm{m}^2\mathrm{s}^{-1}$, the hydrostatic approximation is appropriate to represent motion slower then the Coriolis frequency in realistic simulations of the ocean dynamics. Differences between the two formalisms can be apparent in simulations of specific oceanic processes in an idealized setting.





*Code availability.* No code used

*Data availability.* No data used

*Author contributions.* work done by AW

*Competing interests.* no competing interest

*Acknowledgements.* The work was funded by Shom, contract N 20CP03



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
