# Peer review of "On the hydrostatic approximation in rotating stratified flow"

_EGUsphere, 2024_

## Referee Comment (RC1)

Review of:
On the hydrostatic approximation in rotating stratified flow
by Achim Wirth

**Summary and recommendation**   This paper develops a formal analysis of the differences between hydrostatic and non-hydrostatic equations of motion for incompressible rotating stratified fluids. The author's approach is based on projecting the terms in the space of divergence-free vectors by using Fourier transform, which the author used in previous work to construct a non-hydrostatic model. The proposed method is interesting, but highly technical and often hard to follow. Moreover, as the paper appears to focus exclusively on regimes for which the hydrostatic approximation is generally accepted as valid, it is hard for me to understand what the truly new contributions to existing scientific knowledge are. Indeed, it seems to me that the conclusions reached by the author are essentially the same as those that can be obtained much more simply by comparing the dispersion relations for balanced and inertia-gravity wave plane wave solutions in the hydrostatic versus non-hydrostatic regime. I believe that most readers would be more interested in understanding the errors arising from using the hydrostatic approximations in non-hydrostatic regimes, but this remains unaddressed.

In terms of presentation and structure, the paper is very technical and hard to follow, primarily because the author rarely feels the need to explain why he does what he does and to what purpose. The paper is also parsed with numerous typos and small mathematical errors/inaccuracies. While I could follow the derivations up to Section 3, I started to lose the plot at the beginning of Section 4, when the author introduces an extension of Fourier space to 4 dimensions without explaining why this is needed or useful. The abstract's final statement: 'A special emphasis in on unveiling the physical interpretation of the calculations' is enticing, but I suspect that most readers are likely to disagree with the author's definition of 'physical interpretation'.

On the basis of the above and of the specific comments listed below, reaching publishable status will require a very substantial amount of work aimed at: 1) better identifying the scientific goal of the paper and of the gaps in knowledge it aims to fill; 2) simplifying the mathematical derivations and explaining what these aim to achieve and why; 3) discussing the limitations and caveats of the method arising from the real ocean being not triply periodic; 4) explaining precisely what additional physical insights the proposed method brings compared to simply looking at the linear wave dispersion relationships in the hydrostatic versus non-hydrostatic regimes.

**Specific comments**

1. Abstract, lines 5-6. 'The validity of the hydrostatic approximation [...] wave motion to balanced motion' What do you mean by validity here? For internal waves, the hydrostatic approximation causes some error on the frequency, but then what? Even

at high resolution, one cannot really expect to resolve a full wave spectrum. In any case, isn't that kind of conclusion something that can be deduced from existing linear wave theory? My impression is that a non-hydrostatic treatment was only essential for resonantly excited trapped lee waves, which cannot exist in the hydrostatic regime.

2. Lines 15-16: Equilibrium solutions are in the kernel [..] are in the kernel of its adjoint' What does that mean physically? What are readers supposed to make of this?

3. Lines 17-20 'Using the Heisenberg-Gabor limit [...] physical interpretation of the calculations' I don't understand how these conclusions follow align with the work presented, which seems to pertain only to the discussion of regimes generally regarded as hydrostatic. In many places in the paper, the author mentions instances for which the hydrostatic breaks down, but which are not otherwise discussed theoretically. My reading of the paper is that the author essentially concludes that the hydrostatic approximation is appropriate to hydrostatic regimes, and possibly not appropriate to non-hydrostatic regimes, which is a bit tautological. Or did I misunderstand the paper?

4. Lines 42-43. 'The present paper discusses the gray zone [...] using a purely analytical approach'. I don't understand how the concept of gray zone is pertinent here. Please clarify what you mean. Moreover, if you define identifying the gray zone as an objective of the paper, the expectation is that you should come back to it a the end of the paper.

5. Line 51-52: 'Furthermore, the effect of resonances are not included' Why not? Does that mean that such resonances cannot occur in realistic modelling? What is the impact of the hydrostatic approximation on these? If the hydrostatic approximation cannot correctly capture such resonances, doesn't that imply that the hydrostatic approximation is not necessarily valid for realistic modelling, in contradiction to what is stated in the abstract?

6. Lines 68-85. This is incomprehensible at this stage. Can't this be translated in physical terms? Ultimately, the reader would like to understand whether the formalism developed can help go beyond the conclusions derived from linear wave theory. The author could first remind the reader that when considering plane wave solutions of the linearised equations of motion for a f plane and uniform buoyancy frequency, only the dispersion relation for internal gravity wave is affected by the hydrostatic approximation, but not that for balanced motions. The author should first start by summarising the present state of knowledge and explain what he expects to learn from his considerably more complicated formalism. In doing so, it might be useful to mention the case of trapped lee waves as an example of internal waves that exist only in the non-hydrostatic system. In this regard, can the author's more complicated treatment describe trapped lee waves in the non-hydrostatic case? How can we compare the two systems in that case?

7. Line 150. Is this decomposition physical and mathematically well defined (i.e., unique)? Why is it useful or necessary to consider such a decomposition of the pressure? Moreover, why is the treatment of $P_b$ different from that of the other? I.e., why is it not

$$\Delta^2 P_b = \frac{\partial b}{\partial z}$$

To me, this different treatment makes the approach mathematically and physically inconsistent.

8. Equation (11). I don't understand this equation. The projection operator is supposed to be an operator acting on 3D vector, which is not the case of $\nabla \cdot \mathbf{u} = 0$.

9. Section 3. The method appears to filter out all non-trivial solutions of the Laplace equation $\Delta^2 P = 0$, which in practice are generally essential to construct the full solution of the equations of motion. Can the author comment on this? Moreover, the oceans are clearly not triply periodic. Can the author comment on the limitations and caveats that result from making this clearly unrealistic assumption?

10. Lines 212-215. 'In numerical models based on Fourier representation [...] prescribed to the elliptic solved' - This seems misleading to me because the author compares the accuracy for one Fourier mode making up the solution for the full eliptic problem versus the accuracy of the full solution of the ellipic problem. Since the oceans is not triply periodic, I would expect that one still needs to add a solution of the Laplace equation $\Delta^2 P = 0$ to construct the full solution of the elliptic problem when using a Fourier representation. If not, I don't know how the author can avoid the development of the Gibbs phenomenon near boundaries, which can cause a lack of convergence of the full solution near boundaries. To be fair and meaningful, discussion of accuracy should focus on the full solution for both approaches, which would require that the author discusses how he handles topography and boundary conditions in his approach.

11. Section 4. Fourier space extension. I started to get lost at this stage, so gave up on trying to understand what the author is after. It would be helpful if the author could be somewhat more pedagogical in this approach. What may be clear to him may not be as clear to the reader.

12. As a general comment, could the author clarify what is his intended readership? How are the readers supposed to make use of his results? For instance, an important issue in non-hydrostatic codes using an ellpitic solver is preconditioning. Can the author's results be potentially be used to understand how to improve on existing preconditioning techniques?

---

## Referee Comment (RC3)

**Report on**
**On the hydrostatic approximation**
**in rotating stratified flow**
**by Achim Wirth**

This paper analyses the validity of the hydrostatic approximation widely used in realistic ocean simulation models, to the Naviers-Stokes equations taken as a reference. Sections 2 to 6 set out the mathematical methodology for the comparison based on projection evolution operator in Fourier space and linearised equation. Section 7 presents the error of the approximation under different forcing. Finally, in section 8, the author compares the uncertainty of the approximation with other uncertainties encountered in simulations of ocean dynamics and in the context of the Strait of Gibraltar (friction effect, Doppler). It also quantifies this approximation over a fine wave observation time. The author shows that, overall, the 2 formalisms give approximately the same results, provided that the vertical diffusion is high, which is of the order of magnitude typically used in simulations of ocean dynamics actually. Nevertheless, the difference between the 2 formalisms becomes apparent as the resolution increases and the diffusion models become finer (viscosity will have decreased). Section 8.3 is for me the most important because it implies.

I highly recommend this article, because as the author points out, with the improvement of space-time resolutions in the near future, the differences between two formalism should become apparent and this study will be even more valuable. However, to be honest, I found the text difficult to read, either because of the lack of information that you have to guess at, or because of the links between the arguments that you have to reconstruct. The author should have made it easier to read by presenting his arguments in a more orderly fashion.

I suggest that the result of **this studies would be appropriate for publication** if the authors revised the manuscript taking into account the comments below.

A) **Major comments:**

1. The Navier Stokes equations with the Boussinesq hypothesis have been presented in (1) and (2). However, it is not clear which equation in physical space was used to deduce equation (16) in Fourier space, /which relates to the acceleration of buoyancy $\hat{b}$. From (16) we deduce that:

$$\partial_t b = -(\mathbf{u}.\nabla)b - N^2.u_z + \kappa.\Delta b \tag{R1}$$

This equation can be presented after the discussion of the term linked to $\rho$ from line (133).

2. The original set of hydrostatic equations is not presented, even though author use them in his calculations, so it is not possible to find his results. They are not clearly written in the text, nor are the hypotheses that allow them to be written on the basis

of Navier-Stokes and the Boussinesq hypothesis. From the linearised equations (29) & (32) written by the author, if I have understood correctly, i deduce the original equation as follows:

$$\partial_t u_x = -((\mathbf{u}? \text{ or } \mathbf{u_H}?).\nabla)u_x + f.u_y - \partial_x P + \nu.\Delta u_x \qquad (R2)$$

$$\partial_t u_y = -((\mathbf{u}. \text{ or } \mathbf{u_H}?).\nabla)u_y + -f.u_x - \partial_x P + \nu.\Delta u_x \qquad (R3)$$

$$0 = -\partial_z P + b \qquad (R4)$$

$$\partial_t b = -(\mathbf{u}.\nabla)b - N^2.u_z + \kappa.\Delta b \qquad (R5)$$

$$\partial_z u_z = -\partial_x u_x + \partial_y u_y \qquad (R6)$$

These equations (R2) to (R6) could be written after line (168). In addition, these equations lead to the definition of the operator $\mathbb{P}^H$ in physical space. Once these equations are clearly written in physical space, it will be possible to introduce the equation (R2) to (R6) in Fourier space which is equivalent to equations (13) (14) (15) (16). Also, for clarity, since the central tool in this article is the projection operator $\mathbb{P}$, this operator should be renamed $\mathbb{P}^N$ for the Navier-Stokes equation in (10). From these definitions, it becomes clearer to interpret geometrically the 2 operators $\mathbb{P}^N$ and $\mathbb{P}^H$ which are two operators of projection perpendicular to $\mathbf{k}$ (not clearly expressed in term of geometry in this article) but, the major difference between the two operators, it is that in the case of $\mathbb{P}^H$ one uses the incompressibility to deduce the third component whereas in the case of $\mathbb{P}^H$ the 3 components are used. Following this fundamental distinction, which should be discussed in the text before line 208.

3. Then for the T2 extension, I think that the central motivation is to be able to obtain a difference operator $\mathbb{A}$ between the two formulations $\mathbb{P}^N$ and $\mathbb{P}^H$ which is defined in (22). However, the problem comes from the definition of two different vectors $\mathbf{a}^N$ and $\mathbf{a}^H$, so we would have to write the input of $\mathbb{A}$ using a single vector $\tilde{\mathbf{a}}$ which is defined in (22), i.e. we need to find a common base. Using $\tilde{\mathbf{a}}$ defined in (22) there is only one common vector and it is possible to define in (27) the difference operator between the 2 projections $\mathbb{A}(\mathbf{a}) = \mathbb{P}^H(\mathbf{a}) - \mathbb{P}^N(\mathbf{a})$ and not have to work with $\mathbf{a}^H$ and $\mathbf{a}^N$ which cannot be applied. This point should be made clearer at the start of section 4, otherwise we won't understand the goal of the T2 extension.

4. From this clarification, the T3 extension becomes natural because it consists of writing the evolution equation (linearised) in which we must necessarily distinguish $\hat{b}$ and $\dot{\hat{b}}$ because the 2 formulations treat $\hat{b}$ and $\dot{\hat{b}}$ differently. If this point is detailed at the beginning of section 5, we can then understand the use of (28).

5. line 332: "The validity of the hydrostatic approximation relies also on how strongly the modes $e_1$ , $e_2$ on one side and $e_3$ on the other". I don't understand this sentence can you detailed this point on text ?

6. in section 6.2: why are restricted the space to $D$ ? is it link to my previous question 5. ?

7. For section 8, there is generally no link with the previous study. It is not clear how the author uses the previous mathematical results to deduce physical result:

   i. "I first note that the amplitude of the balanced mode is bounded by ..." : from which section and which formula can this assertion be deduced?

   ii. section 8.2 : "The difference in the horizontal wave velocity, $c$, between the two formalisms is ...", why $\Delta c = \gamma.c$ and $c = \omega/k_h$ ? there is a mathematical formula in your article ? can you detailed the explanation in text.

iii. section 8.3: why "The difference in frequency between the two formalisms is $\sigma_f = \omega\gamma/2$" ? this is not obvious, in particular because $\omega^H = (1 + O(\gamma^2))\omega$. How do you link root mean square $\sigma_f$ of difference between formalism with $\omega$ ? there is a mathematical formula in you article ? if yes which line ? Can you detailed this in text ?

iv. What are the formula used to deduce (79) and (80) ? Give details in the text.

v. "Using the Heisenberg-Gabor limit means that the observation length has to exceed $\gamma^{-2}/4\pi$..." : it is not obvious, can you detailed this point in text.

B) **Minor comments**:

1. equation (12) is not only valid for $a$ but for any function $f$, $(f = a_x, a_y, ux.u_y, ...)$

2. In order to understand that it is difficult to obtain results on $\mathbb{P}$ from physical space, whereas it becomes more obvious in Fourier space, as the author has well written, the author may recall that the operator $\mathbb{P}$ in physical space is written:

$$\mathbb{P}(\mathbf{a}) = \mathbf{a} - \frac{1}{4\pi} \int \int \int \frac{\mathbf{x} - \mathbf{y}}{|\mathbf{x} - \mathbf{y}|^3} \nabla.\mathbf{a} \, d\mathbf{y} \tag{R7}$$

3. For the sake of clarity for all readers, from line 192 onwards, the author should mention a key point which is that the incompressibility condition becomes $i.\mathbf{k}.\mathbf{u} = 0$ in Fourier space that give a geometrical interpretation $\mathbf{k} \perp \mathbf{u}$, and so the operator $\mathbb{P}^N$ and $\mathbb{P}^H$ in Fourier space becomes an operator which projects a second member into a plane perpendicular to $\mathbf{k}$. Moreover, when $kz = 0$ in (20), it should be pointed out that in this case $\mathbb{P}^N = \mathbb{P}^H$. All this geometrical interpretation also help in understanding Figure 3.

4. Figures 2 and 3 should be reversed because figure 3 is used before figure 2 in the text from the projection operator $\mathbb{P}$ in section 2. In fact, contrary to what is written on lines 175 and 217, (T2) and (T3) are not visible in figure 2 and this figure 2 is useful for illustrating what is said in section 6.

5. In equation (4), the various terms $\mathbf{a_u}$, $\mathbf{a_b}$ and $\mathbf{a_f}$ are not defined. Once these terms have been introduced, the equations in (6) ... (10) can be written using these terms, to make them uniform, compact and focused on pressure handling.

6. there is no $\mathbb{P}$ in (11)

7. Figure 3: the points representing $\hat{\mathbf{u}}$ are missing. If we can see the $\mathbf{k}$ vector (which is a good thing), the $(k_z, -k_x)$ vector is not good in terms of scale.

8. line 94: approxiamation.

9. Line 247 : a link should be made with the original hydrostatic equations.

10. line 268 : what is the norm $\sqrt{[\mathbb{A}(\tilde{\mathbf{a}}]^2}$ ? must to define

11. what is $\tilde{\gamma}^\square$ in (35) ? must be defined in (35) ($\tilde{\gamma}^H$ and $\tilde{\gamma} = 1$).

12. could you justify substitution (35). There is a common factor or a linear combination between $\tilde{e}_1, \tilde{e}_2$ and $\tilde{e}^\square_{1,2}$ ?

13. line 333 : square route.

14. line 317 : justify and discuss thermal-wind balance, it is not obvious.

15. what is link between $F_d, F_r$ in (44) and $F = (F_x, F_y, F_z, 0, B)$ ? detail this in text.

16. to make it easier to read, all operators $\mathbb{L}, \mathbb{P}, ...$ should have the index $N$ for Navier-Stokes instead of nothing

17. according your notation, $\mathbf{x} = \mathbf{x}_D$ in (50) and (51) because (51) and (48) is same. Simplify or discuss the difference.

18. what is interpretation $c$ and $s$ in (64) ? is-it group velocity of error ? detailed in text.

19. Section 7.3.1 : there is no link between $\mathbb{G}$ and absolute error previously defined in (49). Must to invert (67) and (65)-(66).

20. to make reading easier, the ratio deifni ligne 205 $\gamma = \sqrt{k_h^2/k_z^2}$ must be highlighted by a line break, a label (eqnarray) and the hypothesis $\gamma \ll 1$ to differentiate it from the one used in the introduction

21. to make it easier to read, the result line 310 $\omega^H \simeq (1 + O(\gamma^2)).\omega$ must be highlighted by a line break, a label (eqnarray) because it is used after.

---

## Author Comment (AC1)

Answers to the editor and the reviewers:

Dear Editor,

I am grateful to the 3 reviewers for their thorough review, corrections and comments as they have increased the quality of the paper. The paper is highly technical and it is not entirely clear to me, how to present its results in a concise and comprehensible form. I have invested many months in the writing alone, changing the presentation several times. I have to admit that the presentation is to the best of my ability and willingness to simplify beyond hiding essential features and will not largely improve in the future. Please find my detailed answers and corrections to the reviewers' comments (reproduced in black) below, written in blue. The corrections performed to the manuscript are given in red and an updated version of the manuscript with the corrections highlighted in red is provided.

Sincerely,
Achim Wirth

Reviewer 1:

Review of:

On the hydrostatic approximation in rotating stratified flow

by Achim Wirth

**Summary and recommendation** This paper develops a formal analysis of the differences between hydrostatic and non-hydrostatic equations of motion for incompressible rotating stratified fluids. The author's approach is based on projecting the terms in the space of divergence-free vectors by using Fourier transform, which the author used in previous work to construct a non-hydrostatic model. The proposed method is interesting, but highly technical and often hard to follow. Moreover, as the paper appears to focus exclusively on regimes for which the hydrostatic approximation is generally accepted as valid, it is hard for me to understand what the truly new contributions to existing scientific knowledge are.

With increasing observational data and computer power, an approximation that was considered valid in the past might no-longer be in the present or futur. That is why more precise tools are necessary to evalute validity. I now added to the beginning of the abstract:

Introducing a Fourier-space projection method and using the Heisenberg-Gabor limit, a formalism is developed to systematically evaluate the role of flatness, stratification, rotation and friction for the fidelity of the hydrostatic approximation.

The introduction now contains the sentence:

In the present work I develop a frame work that allows for discussing the fidelity of the hydrostatic approximation as a function of the flatness $\gamma$, stratification $N$, rotation $f$ and friction $\nu$. The influence of these parameters

revealed by the calculations given in the text, unveil the physics of the hydrostatic approximation.

Indeed, it seems to me that the conclusions reached by the author are essentially the same as those that can be obtained much more simply by comparing the dispersion relations for balanced and inertia-gravity wave plane wave solutions in the hydrostatic versus non-hydrostatic regime. I believe that most readers would be more interested in understanding the errors arising from using the hydrostatic approximations in non-hydrostatic regimes, but this remains unaddressed.

There is no clear-cut distinction between hydrostatic and non-hydrostatic regimes. To investigate the grey-zone between the two sophisticated tools have to be developed and employed. The projection method is well known for the Navier-Stokes formalism, but new (to the best of my knowledge) for the hydrostatic approximation, which asks for an extension to 4 dimensions. This allows for an unified treatment of the Navier-Stokes and the hydrostatic approximation's projections. This is also new (to the best of my knowledge). I now added to the abstract:

The former is well known, while the latter, developed here, asks for an extension of the dynamical space to four dimensions.

The Heisenberg-Gabor-limit provides a mathematical tool and has never been used to evaluate the hydrostatic approximation.

In the papers introduction I write: "These scaling arguments do not include the Coriolis parameter $f$, which is known to further suppress vertical accelerations, this can be seen in the prominent Taylor-Proudman-Poincaré theorem (see i.e. [4])." and "In the present work I develop a frame work that allows for discussing the fidelity of the hydrostatic approximation and offers an understanding of the underlying physics. Calculations are detailed in the text as they unveil the physics."

In the papers conclusions I give my judgment of the situation and the limits of the theory presented: "There are two important points one has to keep in mind: first the hydrostatic approximation is a singular perturbation of the full Navier-Stokes equations, as the highest (temporal) derivative in the equation for the vertical velocity is omitted and expansions in small parameters are no guarantee for the smallness of the error (see i.e. [1]). Second, the hydrostatic approximation breaks down where neither: rotation nor stratification nor scale-ratio constrain the fluid, that is, in fast moving unstratified fluid dynamics. Where, when and how such areas are spontaneously created by the turbulent forced dynamics or external forcing, depends on the situation considered. For these reasons dedicated numerical experiments of an oceanic phenomena have to be employed to determine the fidelity of the hydrostatic approximation for the scales and phenomena studied."

In terms of presentation and structure, the paper is very technical and hard to follow, primarily because the author rarely feels the need to explain

why he does what he does and to what purpose. The paper is also parsed with numerous typos and small mathematical errors/inaccuracies. While I could follow the derivations up to Section 3, I started to lose the plot at the beginning of Section 4, when the author introduces an extension of Fourier space to 4 dimensions without explaining why this is needed or useful.

I added at the beginning of Section 4:

It allows to use the same mathematical formalism, based on 4 dimensional linear algebra, for both formalisms. Only the entries of the matrices differ between the two formalisms. This is necessary as the the two parts of the vertical acceleration, dynamical and buoyancy are treated differently between the formalisms.

The abstract's final statement: 'A special emphasis in on unveiling the physical interpretation of the calculations' is enticing, but I suspect that most readers are likely to disagree with the author's definition of 'physical interpretation'. On the basis of the above and of the specific comments listed below, reaching publishable status will require a very substantial amount of work aimed at: 1) better identifying the scientific goal of the paper and of the gaps in knowledge it aims to fill; 2) simplifying the mathematical derivations and explaining what these aim to achieve and why;

The mathematical formalism, although involved, is only linear algebra. I have spent a long time thinking how to present in the most concise way and I would not know how to improve. I now added in the abstract:

Introducing a Fourier-space projection method and using the Heisenberg-Gabor limit, a formalism is developed to systematically evaluate the role of flatness, stratification, rotation and friction for the fidelity of the hydrostatic approximation.

3) discussing the limitations and caveats of the method arising from the real ocean being not triply periodic;

Although, the ocean is not triply periodic as are other applications of Fourier methods, these methods can be employed in the pseudo-spectral method. the different Fourier modes then interact through the boundary conditions. This is explained in detail in [3]. The last sentence of Section "2 Pressure" is: "The dilemma with the Fourier space is that boundary conditions, which are local in physical space become non-local in Fourier space, see [3] for a detailed discussion and application to ocean modeling". Going beyond this explanation would make the manuscript even more involved.

4) explaining precisely what additional physical insights the proposed method brings compared to simply looking at the linear wave dispersion relationships in the hydrostatic versus non-hydrostatic regimes.

My work shows that the Fourier-space projection used in for the Navier-Stokes formalism can be extended to the hydrostatic approximation when the dimension is extended from 3 to 4 dimensions. This introduces a framework for a systematic comparison between the Navier-Stokes and the hydrostatic approximation. The proposed method shows that the eigen spaces of

the wave dynamics and the balanced dynamics are identical, it includes the effect of rotation it introduces the Heisenberg Gabor limit in the evaluation of the hydrostatic approximation. I do not, however, write every time, that it is new what I do. I did several changes to emphasize the innovative points of my work (see below in my answers to the reviewers).

**Specific comments**

1. Abstract, lines 5-6. 'The validity of the hydrostatic approximation [...] wave motion to balanced motion' What do you mean by validity here? For internal waves, the hydrostatic approximation causes some error on the frequency, but then what? Even at high resolution, one cannot really expect to resolve a full wave spectrum. In any case, isn't that kind of conclusion something that can be deduced from existing linear wave theory? My impression is that a non-hydrostatic treatment was only essential for resonantly excited trapped lee waves, which cannot exist in the hydrostatic regime.

I agree with the reviewer, but for balanced motion the hydrostatic approximation is correct while for motion which involves wave-modes (not necessarily waves) there is some error as the reviewer points out. So I just want to say that the fidelity of the hydrostatic approximation depends not only on the scale ratio $\gamma$ but also on which type of motion, which modes, are considered. This is written in a very condensed way in the abstract but then explained in the text. The sentence is now changed:

 fidelity

2. Lines 15-16: Equilibrium solutions are in the kernel [..] are in the kernel of its adjoint' What does that mean physically? What are readers supposed to make of this?

The sentence is now changed:

 Balanced dynamics is in the kernel of the linear projection-evolution operator and conservation of potential vorticity is expressed by the kernel of its adjoint.

3. Lines 17-20 'Using the Heisenberg-Gabor limit [...] physical interpretation of the calculations' I don't understand how these conclusions follow align with the work presented, which seems to pertain only to the discussion of regimes generally regarded as hydrostatic. In many places in the paper, the author mentions instances for which the hydrostatic breaks down, but which are not otherwise discussed theoretically. My reading of the paper is that the author essentially concludes that the hydrostatic approximation is appropriate to hydrostatic regimes, and possibly not appropriate to non-hydrostatic regimes, which is a bit tautological. Or did I misunderstand the paper?

The essence of the paper is, that it presents a mathematical formalism to investigate fidelity of the hydrostatic approximation. The formalism is based on the eigen modes of the linear projection-evolution operator, which have a clear physical meaning. To my understanding this is new.

The operator theory exposed not only derives the scaling of the error but also provides the prefactors. The Heisenberg-Gabor limit not only allows to compare errors due to the hydrostatic approximation to other sources of error (resolution, internal variability, representation of topography), but also presents a mathematical tool to determine if this errors are significant in a finite size-and-time observation.

4. Lines 42-43. 'The present paper discusses the gray zone [...] using a purely analytical approach'. I don't understand how the concept of gray zone is pertinent here. Please clarify what you mean. Moreover, if you define identifying the gray zone as an objective of the paper, the expectation is that you should come back to it a the end of the paper.

In numerical modeling the "gray zone" usually refers to the scales between eddy resolving models, which do not need a premeditation of eddy fluxes (Gent-McWilliams or other) and coarse non-eddy resolving models that need the parameterization. The (different) gray zone I refer to, is the zone from the large scale processes ($\approx$ 100km), for which the hydrostatic approximation is still valid, but starts to deteriorate for the dynamics of some process at smaller scales, down to the small scale ($\approx$ 100 m) where the hydrostatic approximation clearly fails. This gray zone, which spans around 3 orders of magnitude is at the core of today's oceanographic research. It is to my understanding not a coincidence that this gray zone corresponds closely to the "eddy-gray-zone" (not discussed in the paper). I added in the introduction:

For slow and large-scale dynamics the hydrostatic approximation is valid, for fast small-scale dynamics it is not.

In the conclusions I added:

The calculations in the previous section also show that the Heisenberg-Gabor limit allows to determine the gray zone, for an oceanic process, between the large scales where the hydrostatic approximation is valid, down to the scale where it clearly fails.

5. Line 51-52: 'Furthermore, the effect of resonances are not included' Why not? Does that mean that such resonances cannot occur in realistic modeling? What is the impact of the hydrostatic approximation on these? If the hydrostatic approximation cannot correctly capture such resonances, doesn't that imply that the hydrostatic approximation is not necessarily valid for realistic modeling, in contradiction to what is stated in the abstract?

I argue in the paper that when the viscosity is high enough to parameterize the Ekman layer dynamics, which it must be as resolving Langumuire dynamics asks for resolutions below the 1 m scale, the difference in the resonant wave length are damped. The sentence right before section 7.3.1 was and is: "Resonances are efficiently damped by viscosity if for a forcing at the resonant frequency, $\zeta^2 = \omega^2$, the viscous time-scale is larger than the resonance period, $\tilde{\nu}^2 > \omega^{2}$". And at the end of section 7.3.1 : "If the forcing occurs at the wave frequency of either formalism, resonances or near

resonances dominate the linear dynamics. The hydrostatic approximation breaks down as $\omega^H \neq \omega$ if $k_h \neq 0$ and a resonance occurs in one formalism but not in the other. During resonances or close resonances the scaling is still given by Eq. **??** but the prefactor diverges. Note that if $\tilde{\nu}^2 \gg (\tilde{\gamma}^\square \omega)^2$ resonances are efficiently damped". Further down the entire section "8.1 Masking of Resonance Differences by Friction" is devoted to this point.

6. Lines 68-85. This is incomprehensible at this stage. Can't this be translated in physical terms? Ultimately, the reader would like to understand whether the formalism developed can help go beyond the conclusions derived from linear wave theory. The author could first remind the reader that when considering plane wave solutions of the linearized equations of motion for a f plane and uniform buoyancy frequency, only the dispersion relation for internal gravity wave is affected by the hydrostatic approximation, but not that for balanced motions. The author should first start by summarising the present state of knowledge and explain what he expects to learn from his considerably more complicated formalism. In doing so, it might be useful to mention the case of trapped lee waves as an example of internal waves that exist only in the non-hydrostatic system. In this regard, can the author's more complicated treatment describe trapped lee waves in the non-hydrostatic case? How can we compare the two systems in that case?

Lines 68-85 and beyond, at the end of the introduction expose briefly the plan of the paper, as it is usually done in scientific publications. I do not feel comfortable to talk about the achievements of the new method in the introduction, but have this discussed in the conclusions, where the reader can have a judgement based on the content of the paper. Part of the present state of the knowledge is discussed above. For a subject as the hydrostatic approximation, this discussion can only be incomplete.

The formalism developed presents a systematic tool to use linear wave theory, but does not go beyond it, as it discusses the problem for each wavevector and forcing frequency independently. This is made clear in several places in the paper. Extending the formalism to trapped lee-waves and exposing the difference between the formlisms asks for a dedicated paper. In this case specified back ground shear flows and stratifications have to be employed, which (as I suspect) can not be represented by a single mode but asks for their superposition, which asks for dedicated calculations.

A counter example are, however, trapped lee-waves generated by topography, of which the dynamics differs qualitatively, when the hydrostatic approximation is employed, as discussed by [5]

7. Line 150. Is this decomposition physical and mathematically well defined (i.e., unique)? Why is it useful or necessary to consider such a decomposition of the pressure? Moreover, why is the treatment of $P_b$ different from that of the other? I.e., why is it not $\Delta^2 P_b = \frac{\partial b}{\partial z}$ To me, this different treatment makes the approach mathematically and physically inconsistent.

Eq. 8 is now corrected (in a previous version I had distinguished also between the background buoyancy and the variation to it with gives rise to the available potential energy, such distinction (although very instructive) does not add to the understanding for the here presented formalism):

The third term is due to buoyancy:

$$\nabla^2 P_b = \nabla \cdot b. \tag{1}$$

Calculating the pressures above asks for solving an elliptic (non-local) problem. It is instructive to distinguish the different terms as the vertical acceleration of the of the r.h.s of Eqs. **??**, only, are neglected in the hydrostatic approximation.

The distinction is necessary as in the hydrostatic approximation the dynamic pressure and the buoyant pressure are treated differently.

8. Equation (11). I don't understand this equation. The projection operator is supposed to be an operator acting on 3D vector, which is not the case of $\nabla \cdot u = 0$.

Eq. 11 is now corrected.

$$\nabla \cdot \mathbb{P}(\mathbf{u}) = 0.$$

9. Section 3. The method appears to filter out all non-trivial solutions of the Laplace equation $\Delta^2 P = 0$, which in practice are generally essential to construct the full solution of the equations of motion. Can the author comment on this? Moreover, the oceans are clearly not triply periodic. Can the author comment on the limitations and caveats that result from making this clearly unrealistic assumption?

I do not understand the comment about the Laplace equation: $\Delta P = \nabla^2 P = 0$. Although, the ocean is not triply periodic as are other applications of Fourier methods, these methods can be employed in the pseudo-spectral method. the different Fourier modes then interact through the boundary conditions. This is explained in detail in [3]. The last sentence of Section "2 Pressure" is: "The dilemma with the Fourier space is that boundary conditions, which are local in physical space become non-local in Fourier space, see [3] for a detailed discussion and application to ocean modeling". Going beyond this explanation would make the manuscript even more involved.

10. Lines 212-215. 'In numerical models based on Fourier representation [...] prescribed to the elliptic solved' - This seems misleading to me because the author compares the accuracy for one Fourier mode making up the solution for the full elliptic problem versus the accuracy of the full solution of the elliptic problem. Since the oceans is not triply periodic, I would expect that one still needs to add a solution of the Laplace equation $\Delta^2 P = 0$ to construct the full solution of the elliptic problem when using a Fourier representation. If not, I don't know how the author can avoid the

development of the Gibbs phenomenon near boundaries, which can cause a lack of convergence of the full solution near boundaries. To be fair and meaningful, discussion of accuracy should focus on the full solution for both approaches, which would require that the author discusses how he handles topography and boundary conditions in his approach.

The reviewer is right and points to an important problem, which I discussed in detail in [3]. In triply periodic problems Fourier methods satisfy the incompressibility condition at machine precision. When boundary conditions have to be satisfied the situation is more complicated, as boundary conditions are local in physical space, whereas the incompressibility condition is local in Fourier space and it is therefore difficult to impose both simultaneously. This and the question of how to avoid Gibbs oscillations near the boundary are discussed in [3], where it is also suggested to solve a two dimensional elliptic problem in Fourier space to satisfy boundary conditions rather than solving a three dimensional elliptic problem in physical space to satisfy the incompressibility condition. I now added to the manuscript:

The question of how to employ Fourier methods in ocean modeling, where boundary conditions have to be satisfied, is discussed in detail in [3].

11. Section 4. Fourier space extension. I started to get lost at this stage, so gave up on trying to understand what the author is after. It would be helpful if the author could be somewhat more pedagogical in this approach. What may be clear to him may not be as clear to the reader.

The paper is highly technical, but to my understanding this is unavoidable. It presents a mathematical framework that allows for systematically evaluating the hydrostatic approximation. To this end I introduce a 4-dimensional matrix representation, in which the non-hydrostatic and hydrostatic formalisms can both be written. I added in the beginning of Section 4:

It allows to use the same mathematical formalism, based on 4 dimensional linear algebra, for both formalisms. Only the entries of the matrices differ between the two formalisms.This is necessary as the the two parts of the vertical acceleration, dynamical and buoyancy are treated differently between the formalisms.

12. As a general comment, could the author clarify what is his intended readership? How are the readers supposed to make use of his results? For instance, an important issue in non-hydrostatic codes using an elliptic solver is preconditioning. Can the author's results be potentially be used to understand how to improve on existing preconditioning techniques?

Hydrostatic models are used for a variety of applications, they are the "workhorses" of climate models, as said in the first sentence of the abstract. The paper tries to discuss the hydrostatic approximation in a broader mathematical context. It gives concrete examples how the developed theory can be used to determine the validity of the hydrostatic-approximation based on

the Heisenberg-Gabor limit.

I added in the introduction:

To accelerate the iteratively solved elliptic problem of finding the non-hydrostatic pressure, it is often preconditioned by the more easily obtained hydrostatic pressure.

I agree with the reviewer, that an increased understanding of the difference of the two formalisms can potentially lead to improve the preconditioning, but I do not want to speculate on this point.

Reviewer 2:

I thank the reviewer for his comments as they have helped to increase the quality of the paper.

This paper deals with the validity of the hydrostatic approximation generally adopted in numerical simulations of ocean dynamics, in comparison to the resolution of the fully 3D Navier-Stokes (NS) equations, in the presence of rotation and stratification effects. The theoretical basis for such an approximation relies on the fact that the typical scale of vertical processes, say $H$, is much smaller than that of horizontal ones, say $L : \gamma = H/L$ is much smaller than one.

The goal of the paper is to quantify, on a scale-by-scale/frequency-by-frequency basis, the error that is associated with the hydrostatic approximation: the hydrostatic vs NS linearized dynamics are discussed in Fourier space and the error is quantified power of $\gamma$.

I find the paper interesting and worth of publication. However it is not an easy reading, and I think that its impact could be increased by improving the notation and the physical explanations. In the following, I try and share some considerations in the above direction. My evaluation is thus: Revise and Accept.

————————————————————————-

0) Generally speaking, the abstract and the introduction section should be more explicit on the main results of the paper. What do we learn? Are the results of interest only for people doing ocean numerical modeling, or a more general lesson can be learnt? The discussion is of course on the linear dynamics, but we know that non linear terms are crucial: how these impact the validity of hydrostatic approximation?

The main point of the paper is, that it presents a new formalism that allows to consider the Navier-Stokes equations and their hydrostatic approximation in an unifying approach. And it therefore allows to evaluate the latter. I now added to the abstract (second sentence):

Introducing a Fourier-space projection method and using the Heisenberg-Gabor limit, a formalism is developed to systematically evaluate the hydrostatic approximation.

In the introduction was and is written:

"In the present work I develop a frame work that allows for discussing the fidelity of the hydrostatic approximation and offers an understanding of the underlying physics."

Which describes the purpose of the paper.

I prefer to talk about the limitations of my theory in the conclusions, when the reader can judge. There is and was the following paragraph in the manuscript:

"There are two important points one has to keep in mind: first the hydrostatic approximation is a singular perturbation of the full Navier-Stokes equations, as the highest (temporal) derivative in the equation for the vertical velocity is omitted and expansions in small parameters are no guarantee for the smallness of the error (see i.e. [1]). Second, the hydrostatic approximation breaks down where neither: rotation nor stratification nor scale-ratio constrain the fluid, that is, in turbulent unstratified non-rotating fluid dynamics. Where, when and how such areas are spontaneously created by the turbulent dynamics or external forcing, depends on the situation considered. For these reasons dedicated numerical experiments of an oceanic phenomena have to be employed to determine the fidelity of the hydrostatic approximation for the scales and phenomena studied."

1) It seems to me that he content of Sections 3-6 is well known, and is a little bit tedious. I would probably move details to an Appendix and discuss the order of the error in terms of $\gamma$, and possible applications.

The reviewer is right the paper is technical and tedious. Section 3 is known to a small part of physical oceanographers familiar with Fourier-spectral-methods. Section 4, the increase of dimensions is, to the best of my knowledge, new. Section 5 and 6 are the application of section 4 and have (again, to the best of my knowledge) never been discussed. I have considered the option of putting the explicit forms of the vectors and matrices in an appendix, but they present the essence of the exposed theory and the mathematics used is just basic linear algebra in 4 or 5 dimensions. In fact, giving the explicit vectors and matrices, rather than showing only the symbols in the text (and putting the explicit form in appendix), makes the manuscript more concrete and understandable. It was written in the text (line 57): "Calculations are detailed in the text as they unveil the physics." And further down (line 65):"This leads to a formal frame-work to investigate the validity of the hydrostatic approximation using linear algebra in low ($\leq 5$) dimensional spaces."

The projection method is well known for the Navier-Stokes formalism, but new (to the best of my knowledge) for the hydrostatic approximation, which asks for an extension to 4 dimensions. This allows for a unified treatment of the Navier-Stokes and the hydrostatic approximation's projections. This is also new (to the best of my knowledge). I now added to the abstract:

The former is well known, while the latter, developed here, asks for an extension of the dynamical space to four dimensions.

2) it would be useful to state from the introduction the range of variation of the parameter $\gamma$ in the ocean dynamics applications, and its typical values.

I now added to the Introduction:

For the basin-wide circulation $\gamma \approx 10^{-4}$ for a mesoscale eddy $\gamma \approx 10^{-2}$ and their dynamics is well described by the hydrostatic models. They acceptably model gravity currents, coastal currents and internal inertial-waves of large wavelength, with $\gamma \approx .01 - .1$, but are inappropriate to integrate shorter internal waves $\gamma \approx .1 - 1$ or the mixed layer dynamics and three dimensional mixing, where $\gamma \approx 1$.

3) the notation is not always reader-friendly. For example, the symbol "tilde" is used to redefine variables but not with the same meaning, which results in some confusion. At line 185, it is used to define the inverse friction time $\nu k^2$ while at line 220 it is used to define the four-components vector $(\hat{a_x}, (\hat{a_y}(\hat{a_z}(\hat{a_b}))$. So the tilde has not a unique meaning. It is better to avoid such confusion.

The symbol: $\tilde{\nu}$ is now replaced by: $\breve{\nu}$.

4) In figure 3, page 8, it is used the notation with "the symbol $\square$ " that is however introduced only at line 281 at page 12, "the symbol $\square$ is a place holder for " " for the Navier-Stokes or "H" for the hydrostatic formalism. " The best is probably to avoid it, and use "NS", "H" and "NS,H" when NS, hydrostatic or both apply, respectively.

The superscript "N" is now used for Navie-Stokes and $\square$ is kept when it applies to both formalism. This naming convention is now reminded in the legend of Fig. 2

5) It is unclear what the author mean with "(real-component) Fourier sub-space" in the caption of Figure 3, by saying "Schematic view of the (real-component) Fourier subspace spanned by the velocity components...".

Fourier components are complex numbers, so in the 2D schematic only the real-part of the Fourier-coefficients (or the imaginary-part) can be drawn (the full problem is $3 \times 2$ dimensions). That is the coefficients of the cos (or the sin). I now corrected components $\rightarrow$ coefficients and changed:

"Schematic view of the Fourier subspace spanned by the real-part of the Fourier coefficients of the velocity $(\hat{u}_x(k_x, k_z), \hat{u}_z(k_x, k_z))$ for the mode $\mathbf{k} = (k_x, k_z \neq 0)$, given by the thin-red-vector."

6) at line 572, the bottom friction coefficient has no physical dimensions: are these missing?

The bottom friction drag coefficient for a quadratic drag law has no dimension. The explication is now extended to:

At larger horizontal scales friction is parameterised through turbulent bottom friction, which has an inverse friction time of $c_D u/H$ A bottom friction drag coefficient of $c_D \approx 10^{-3}$, a current speed of $u = 10^{-1}$ m s$^{-1}$ and a resonance frequency $\omega \approx 10^{-4}$ s$^{-1}$ shows that bottom friction is sufficient to hide the difference in resonance frequencies $(\gamma\omega)$ only if the ocean layer thickness in meters is below $\gamma^{-1}$

7) at line 573-574, it is written "shows that bottom friction is sufficient to hide the difference in resonance frequencies only if the ocean layer thickness in meters is below $\gamma^{-1}$. Is the author meaning "...in resonance frequencies only if the ocean layer thickness in meters is below $(\gamma \Delta x)^{-1}$?

8) at line 580, it is written that "The difference in the horizontal wave velocity, c, between the two formalisms is $\Delta c = \gamma c$. Is this obvious?

I now corrected:

The square-root of difference in the square of the horizontal wave velocity between the two formalisms, $\Delta c = \sqrt{(c^H)^2 - (c^N)^2}$ is $\Delta c = \gamma c = \gamma \omega^N / k_h = \omega^N / k_z$ (see Eq. **??**), which equals the vertical phase speed.

9) I have a question concerning intermittent burst that are known to characterise turbulent stratified flows. In PHYSICAL REVIEW E 89, 043002 (2014), it is shown that strong intermittent events characterizing temporal dynamics come from the direct coupling between vertical velocity and temperature fluctuations. I wonder if these are relevant for the ocean dynamics, and if yes which is their fate in the hydrostatic approximation.

The paper the reviewer refers to, considers non-linear amplification of Kelvin-Helmholtz dynamics. For such dynamics $\gamma \approx 1$ and it is therefore far from hydrostatic. Please note that in the introduction of my paper I wrote: "At small scales the turbulent motion is fully three-dimensional and the three-dimensional Navier-Stokes equations have to be solved." And in the conclusion: "There are two important points one has to keep in mind: first the hydrostatic approximation is a singular perturbation of the full Navier-Stokes equations, as the highest (temporal) derivative in the equation for the vertical velocity is omitted and expansions in small parameters are no guarantee for the smallness of the error (see i.e. Bender et al. (1999)). Second, the hydrostatic approximation breaks down where neither: rotation nor stratification nor scale-ratio constrain the fluid, that is, in fast moving unstratified fluid dynamics. Where, when and how such areas are spontaneously created by the turbulent forced dynamics or external forcing, depends on the situation considered. For these reasons dedicated numerical experiments of an oceanic phenomena have to be employed to determine the fidelity of the hydrostatic approximation for the scales and phenomena studied". Such processes act on the large scale dynamics, through an increase in the local mixing, and this action has to be parameterized in hydrostatic models.

———

Typos:

line 127 : "from left to right on the l.h.s. of Eq. 3" $---> $ "from left to right on the r.h.s. of Eq. 3"

Done

line 128 : divegenceless $---> $ divergenceless

Done

Citation: https://doi.org/10.5194/egusphere-2024-3307-RC2

Report on

On the hydrostatic approximation in rotating stratified flow

by Achim Wirth

This paper analyses the validity of the hydrostatic approximation widely used in realistic ocean simulation models, to the Naviers-Stokes equations taken as a reference. Sections 2 to 6 set out the mathematical methodology for the comparison based on projection evolution operator in Fourier space and linearized equation. Section 7 presents the error of the approx- imation under different forcing. Finally, in section 8, the author compares the uncertainty of the approximation with other uncertainties encountered in simulations of ocean dynamics and in the context of the Strait of Gibraltar (friction effect, Doppler). It also quantifies this approx- imation over a fine wave observation time. The author shows that, overall, the 2 formalisms give approximately the same results, provided that the vertical diffusion is high, which is of the order of magnitude typically used in simulations of ocean dynamics actually. Nevertheless, the difference between the 2 formalisms becomes apparent as the resolution increases and the diffusion models be- come finer (viscosity will have decreased). Section 8.3 is for me the most important because it implies.

I highly recommend this article, because as the author points out, with the improvement of space-time resolutions in the near future, the differences between two formalism should become apparent and this study will be even more valuable. However, to be honest, I found the text difficult to read, either because of the lack of information that you have to guess at, or because of the links between the arguments that you have to reconstruct. The author should have made it easier to read by presenting his arguments in a more orderly fashion. I suggest that the result of this studies would be appropriate for publication if the authors revised the manuscript taking into account the comments below.

A) Major comments:

1. The Navier Stokes equations with the Boussinesq hypothesis have been presented in (1) and (2). However, it is not clear which equation in physical space was used to deduce equation (16) in Fourier space, /which relates to the acceleration of buoyancy ˆb. From (16) we deduce that:

$$\partial_t b = -(u \cdot \nabla)b - N^2 u_z + \kappa \Delta b \qquad \text{(R1)} \qquad (2)$$

This equation can be presented after the discussion of the term linked to $\rho$ from line (133).

I now added at the location suggest by the reviewer the equation in physical space:

The evolution equation of the buoyancy anomaly is:

$$\partial_t b = -((\mathbf{u} \cdot \nabla)b - u_z N^2 + \kappa \nabla^2 b. \qquad (3)$$

For the integration step we write the evolution equation, Eq. **??**, without the projector and the buoyancy equation, Eq 3, in Fourier space:

2. The original set of hydrostatic equations is not presented, even though author use them in his calculations, so it is not possible to find his results. They are not clearly written in the text, nor are the hypotheses that allow them to be written on the basis of Navier-Stokes and the Boussinesq hypothesis. From the linearized equations (29) & (32) written by the author, if I have understood correctly, i deduce the original equation as follows:

$$\partial_t u_x = -((u?oru_H?).\nabla)u_x + f.u_y - \partial_x P + \nu.\Delta u_x \qquad \text{(R2)}$$

$$\partial_t u_y = -((u.oru_H?).\nabla)u_y + -f.u_x - \partial_x P + \nu.\Delta u_x \qquad \text{(R3)}$$

$$0 = -\partial_z P + b(R4) \qquad \text{(R4)}$$

$$\partial_t b = -(u.\nabla)b - N^2.u_z + \kappa\Delta b \qquad \text{(R5)}$$

$$\partial_z u_z = -\partial_x u_x + \partial_y u_y \qquad \text{(R6)}$$

These equations (R2) to (R6) could be written after line (168). In addition, these equations lead to the definition of the operator $\mathbb{P}^H$ in physical space. Once these equations are clearly written in physical space, it will be possible to introduce the equation (R2) to (R6) in Fourier space which is equivalent to equations (13) (14) (15) (16).

The equation for the evolution step is the same in both formalisms, it is Eq. 1 (the third component is never used in the hydrostatic formalism as the vertical eulerian-acceleration is calculated based on the horizontal divergence). I now added in the beginning of section 3:

For both formalisms the integration starts from a divergence-free initial-condition $\widehat{\mathbf{u}}_i$ using equation Eq. 1.

Also, for clarity, since the central tool in this article is the projection operator $\mathbb{P}$, this operator should be renamed $\mathbb{P}^N$ for the Navier-Stokes equation in (10).

$\mathbb{P}$ is now renamed $\mathbb{P}^N$ as suggested by the reviewer.

From these definitions, it becomes clearer to interpret geometrically the 2 operators $\mathbb{P}^N$ and $\mathbb{P}^H$ which are two operators of projection perpendicular to k (not clearly expressed in term of geometry in this article) but, the major difference between the two operators, it is that in the case of $\mathbb{P}^H$ one uses the incompressibility to deduce the third component whereas in the case of $\mathbb{P}^H$ the 3 components are used.

I disagree with the reviewer $\mathbb{P}^N$ is perpendicular to $\mathbf{k}$, but $\mathbb{P}^H$ is not. It was and is written in the legend of the figure explaining the geometry of the formalism: "The Navier-Stokes projection is orthogonal to the subspace of zero-divergence and ends at $\widehat{\mathbf{u}}$. The hydrostatic projection is in the $\widehat{u}_z$ direction and ends at $\widehat{\mathbf{u}}^H$."

Following this fundamental distinction, which should be discussed in the text before line 208.

I prefer to restrict the geometric interpretation to the legend of the figure, I now added to the text:

and the vertical dynamical acceleration is calculated from its horizontal counterparts. A geometrical interpretation of the two projections and their differences is shown in Fig. **??**.

3. Then for the T2 extension, I think that the central motivation is to be able to obtain a difference operator $\mathbb{A}$ between the two formulations $\mathbb{P}^N$ and $\mathbb{P}^H$ which is defined in (22). However, the problem comes from the definition of two different vectors $\mathbf{a}^N$ and $\mathbf{a}^H$, so we would have to write the input of $\mathbb{A}$ using a single vector $\tilde{\mathbf{a}}$ which is defined in (22), i.e. we need to find a common base. Using $\tilde{\mathbf{a}}$ defined in (22) there is only one common vector and it is possible to define in (27) the difference operator between the 2 projections $\mathbb{A}(\tilde{\mathbf{a}}) = \mathbb{P}^H(\tilde{\mathbf{a}}) - \mathbb{P}^N(\tilde{\mathbf{a}})$ and not have to work with $\mathbf{a}^H$ and $\mathbf{a}^N$ which cannot be applied. This point should be made clearer at the start of section 4, otherwise we won't understand the goal of the T2 extension.

The reviewer is right, there is a difference in the definition of $\mathbb{P}^H$ and its representation in the Figure explaining the projection, because $\mathbb{P}^H$ is not only the projection in the vertical "down" on the subspace of zero divergence (this only the last part), there are two more steps involved: the forgetting of the vertical dynamic acceleration and the addition of the buoyancy acceleration to the horizontal acceleration. The Fig. and its legend are now corrected. In the text it was ok. When I give a talk on the subject I spent 10 min on this fig. as it explains all the projection formalism. I added in the text:

A geometric illustration of the projections, for both formalisms is given in Fig. **??**. The actual operators are presented below.

4. From this clarification, the T3 extension becomes natural because it consists of writing the evolution equation (linearized) in which we must necessarily distinguish $\hat{b}$ and $\dot{\hat{b}}$ because the 2 formulations treat $\hat{b}$ and $\dot{\hat{b}}$ differently. If this point is detailed at the beginning of section 5, we can then understand the use of (28).

I now added to the text (after the former Eq.(28):

Note that the extension to five dimensions is necessary as: firstly, the two formalisms treat the buoyancy and the vertical-dynamic acceleration differently and I want to present both in the same formalism and, secondly, the vertical velocity acts on the time-derivative of the buoyancy through the background stratification.

5. line 332: "The validity of the hydrostatic approximation relies also on how strongly the modes $e_1, e_2$ on one side and $e_3$ on the other". I don't understand this sentence can you detailed this point on text ?

I added the sentence:

If all the energy resides in the third mode, the hydrostatic approximation is exact, while the hydrostatic solution can diverge at a rate $\gamma$, for the energy

in the wave-modes.

6. in section 6.2: why are restricted the space to D ? is it link to my previous question 5. ?

The dynamics happens in $D$ only, as the fourth vector is in the kernel of $\mathbb{L}$. I now changed the first sentence of section 6.2 to:

As the fourth vector, in both formalisms, is in the kernel of the corresponding operator, the problem simplifies when it is restricted to the linear operator in the basis $D = (\tilde{\mathbf{e}}_1, \tilde{\mathbf{e}}_2, \mathbf{e}_3)$, which is common to both operators:

7. For section 8, there is generally no link with the previous study. It is not clear how the author uses the previous mathematical results to deduce physical result:

Section 8.1 is an application of the theory and equations presented in Section 7 and the equations are cited. It is made clear that the discussion depends on the dispersion relation introduced above. I now added in the beginning of the section:

I first note that the amplitude of the balanced mode is bounded by $\max(\breve{\nu}, \zeta)^{-1}$ times the forcing amplitude (see Eq. **??**).

i. "I first note that the amplitude of the balanced mode is bounded by ..." : from which section and which formula can this assertion be deduced?

It is now corrected:

I first note that the amplitude of the balanced mode is bounded by $\max(\breve{\nu}, \zeta)^{-1}$ times the forcing amplitude (see Eq. **??**).

ii. section 8.2 : "The difference in the horizontal wave velocity, c, between the two formalisms is ...", why $\Delta c = \gamma.c$ and $c = \omega/k_h$ ? there is a mathematical formula in your article ? can you detailed the explanation in text.

It is now corrected:

The square-root of difference in the square of the horizontal wave velocity between the two formalisms, $\Delta c = \sqrt{(c^N)^2 - (c^H)^2}$ is $\Delta c = \gamma c = \gamma \omega^N/k_h = \omega^N/k_z$ (see Eq. **??**), which equals the vertical phase speed.

iii. section 8.3: why "The difference in frequency between the two formalisms is $\sigma_f = \omega\gamma/2$" ? this is not obvious, in particular because $\omega^H = (1 + O(\gamma^2))\omega$. How do you link root mean square $\sigma f$ of difference between formalism with $\omega$ ? there is a mathematical formula in you article ? if yes which line ? Can you detailed this in text ?

I now expanded:

The difference in the square frequencies between the two formalisms, obtained from Eq. **??**, is $(\omega^N\gamma)^2$ . For the observations to allow to discriminate between the two frequencies, we need $\sigma_f \leq \omega^N\gamma$, the Heisenberg-Gabor limit leads to an observation time $\sigma_t \geq (2\pi\omega\gamma)^{-1}$.

And:

In the present paragraph I will compare the difference in the frequencies between the two formalisms to the observation error in the horizontal and

the vertical wave number. The calculations are based on the dispersion relations given in Eq. **??**.

iv. What are the formula used to deduce (79) and (80) ? Give details in the text.

I now expanded:

To calculate the difference in wave vectors $\mathbf{k}^H$ and $\mathbf{k}$ between both formalisms for a given wave-frequency, the equation: $\omega^H(\mathbf{k}^H) = \omega^N(\mathbf{k})$ has to be solved, based on Eq. **??**.

v. "Using the Heisenberg-Gabor limit means that the observation length has to exceed $\gamma^{-2}/4\pi$ ..." : it is not obvious, can you detailed this point in text.

If the two wave numbers differ by $\sigma_f = \gamma k$ we need an observation length of $\sigma_t \geq (4\pi\gamma k)^{-1}$ to be able to discriminate between the two frequencies. I now added:

(Eq. **??**)

B) Minor comments:

1. equation (12) is not only valid for a but for any function $f$ , ($f = a_x, a_y, u_x.u_y, ...$)

The sentence is now replaced by:

The Fourier transform of any $L^3-$periodic function $f$ is:

$$\widehat{f}(\mathbf{k}) = \frac{1}{L^3} \int_{L^3} e^{i(k_x x + k_y y + k_z z)} f(x,y,z)\, dx\, dy\, dz, \tag{4}$$

2. In order to understand that it is difficult to obtain results on $\mathbb{P}$ from physical space, whereas it becomes more obvious in Fourier space, as the author has well written, the author may recall that the operator $\mathbb{P}$ in physical space is written:

$$\mathbb{P}(\mathbf{a}) = \mathbf{a} - \frac{1}{4\pi} \int \int \int \frac{\mathbf{x} - \mathbf{y}}{|\mathbf{x} - \mathbf{y}|^3} \nabla.\mathbf{a}\, d\mathbf{y}\, (R7) \tag{5}$$

The reviewer is right, but I prefer not present this formula as (i) there are already too many formulas, (ii) this real-space relation is much more involved than the Fourier-space representation (which is just a matrix multiplication and (iii) the real-space relation is not used in the sequel.

I now added:

In Fourier space it is a matrix multiplication, whereas in real-space it is a (non-local) integration over the whole domain.

3. For the sake of clarity for all readers, from line 192 onwards, the author should mention a key point which is that the incompressibility condition becomes $i.\mathbf{k}.\mathbf{u} = 0$ in Fourier space that give a geometrical interpretation $\mathbf{k} \perp \mathbf{u}$, and so the operator $\mathbb{P}^N$ and $\mathbb{P}^H$ in Fourier space becomes an operator which projects a second member into a plane perpendicular to $\mathbf{k}$. Moreover,

when $\mathbf{k}_z = 0$ in (20), it should be pointed out that in this case $\mathbb{P}^N = \mathbb{P}^H$. All this geometrical interpretation also help in understanding Figure 3.

The sentence is now extended to:

The integration step is followed by a projection on the divergence-free sub-space, which is formed by vector fields perpendicular to the wave-vector, *i.e.* $\mathbb{P}(\widehat{\mathbf{a}}) \cdot \mathbf{k} = 0$.

It was and is written in the manuscript that: "For $k_z = 0$ the hydrostatic projection equals the Navier-Stokes projection applied to the first two components." The vertical velocity / acceleration is formally treated differently between the two projections for $k_z = 0$, but this has no importance as it is vanishing due to horizontal boundaries, which are always present in GFD.

4. Figures 2 and 3 should be reversed because figure 3 is used before figure 2 in the text from the projection operator P in section 2. In fact, contrary to what is written on lines 175 and 217, (T2) and (T3) are not visible in figure 2 and this figure 2 is useful for illustrating what is said in section 6.

Figures 2 and 3 are now reversed and the reference corrected

5. In equation (4), the various terms $\mathbf{a_u}, \mathbf{a_b}$ and $\mathbf{a_f}$ are not defined. Once these terms have been introduced, the equations in (6) ... (10) can be written using these terms, to make them uniform, compact and focused on pressure handling.

6. there is no $\mathbb{P}$ in (11)

Corrected, $\mathbb{P}$ now acts on $\mathbf{u}$.

7. Figure 3: the points representing $\widehat{\mathbf{u}}$ are missing. If we can see the $\mathbf{k}$ vector (which is a good thing), the $(k_z, -k_x)$ vector is not good in terms of scale.

Done

8. line 94: approxiamation.

Done

9. Line 247 : a link should be made with the original hydrostatic equations.

I added:

(thermal-wind balance, see [2])

10. line 268 : what is the norm $\sqrt{[\mathbb{A}(\tilde{\mathbf{a}})]^2}$ ? must to define

Please note that $\mathbb{A}(\tilde{\mathbf{a}})$ is a vector, so it is the usual vector norm (length of the vector).

11. what is $\tilde{\gamma}^\square$ in (35) ? must be defined in (35) ($\tilde{\gamma}^H$ and $\tilde{\gamma} = 1$).

I now added:

and $\tilde{\gamma}^N = 1$.

12. could you justify substitution (35). There is a common factor or a linear combination between $\tilde{\mathbf{e}}_1$, $\tilde{\mathbf{e}}_2$ and $\tilde{\mathbf{e}}_{1,2}^\square$ ?

It was and is written: "[the substituted vectors] of which the directions are independent of the projection", which allows to compare the two formalisms. Without this property the paper would not have been written.

Furthermore the explicit forms of these vectors, one has a vanishing vertical dynamic acceleration and the other a vanishing vertical buoyant acceleration is discussed in detail further below. I prefer not to mention these properties right after the introduction of the vectors as they would perturb the reader (to my understanding).

13. line 333 : square route.

Done

14. line 317 : justify and discuss thermal-wind balance, it is not obvious.

I added on line 247 (see my answer to your point 9) :

(thermal-wind balance, see [2])

15. what is link between $F_d, F_r$ in (44) and $F = (F_x, F_y, F_z, 0, B)$ ? detail this in text.

I now added in section 6.1:

$(\widehat{F}_d = (k_x \widehat{F}_x + k_y \widehat{F}_y)/k_h$ and $\widehat{F}_r = (k_x \widehat{F}_y - k_y \widehat{F}_x)/k_h)$

16. to make it easier to read, all operators $\mathbb{L}, \mathbb{P}, ...$ should have the index N for Navier- Stokes instead of nothing

Done

17. according your notation, $x = x_D$ in (50) and (51) because (51) and (48) is same. Simplify or discuss the difference.

The reviewer is right, it is now $\widehat{\mathbf{x}}$ everywhere.

18. what is interpretation $c$ and $s$ in (64) ? is-it group velocity of error ? detailed in text.

They are just functions, I don't have an easy explanation. If viscosity vanishes $\tilde{s}$ is a sine function and $\tilde{c}$ a cosine function.

19. Section 7.3.1 : there is no link between $\mathbb{G}$ and absolute error previously defined in (49). Must to invert (67) and (65)-(66).

In the (former) Eq. 49 "E" is now replaced by "$\widehat{E}_0$", as it considers the stationary forcing case. I prefer to discuss the special cases (stationary forcing) before the general case. I also prefer to discuss the error in the evolution operator before the total error.

20. to make reading easier, the ratio deifni ligne 205 $\gamma = \sqrt{k_h^2/k_z^2}$ z must be highlighted by a line break, a label (eqnarray) and the hypothesis $\gamma \ll 1$ to differentiate it from the one used in the introduction

Please note that the definition is identical to the one in the introduction, only expressed in wave number, rather than in wave-length. The definition is now highlighted.

$$\gamma = \sqrt{k_h^2/k_z^2}, \tag{6}$$

21. to make it easier to read, the result line 310 $\omega^H \approx (1 + O(\gamma^2))$. $\omega$ must be highlighted by a line break, a label (eqnarray) because it is used after.

I do not understand, $\omega^H$ is given in Eq. 34, now Eq. 36.

**References**

[1] Bender, C. M., Orszag, S., and Orszag, S. A.: Advanced mathematical methods for scientists and engineers I: Asymptotic methods and perturbation theory, vol. 1, Springer Science & Business Media, 1999.

[2] Vallis, G. K.: Atmospheric and oceanic fluid dynamics, Cambridge University Press, 2017.

[3] Wirth, A.: A non-hydrostatic flat-bottom ocean model entirely based on Fourier expansion, Ocean Modelling, 9, 71–87, 2005.

[4] Wirth, A. and Barnier, B.: Mean circulation and structures of tilted ocean deep convection, Journal of physical oceanography, 38, 803–816, 2008.

[5] Yu, C. and Teixeira, M.: Impact of non-hydrostatic effects and trapped lee waves on mountain-wave drag in directionally sheared flow, Quarterly Journal of the Royal Meteorological Society, 141, 1572–1585, 2015.

---

## Referee Report (RR1)

**Report on**
**On the hydrostatic approximation in rotating stratified flow**
**by Achim Wirth**
**Review 2**

The new version of the text presented by the author has been much improved. Nevertheless, there are still grey areas that need to be clarified. I suggest that the result of **this studies would be appropriate for publication** if the author revised the manuscript taking into account the comments below. I am confident that author will carefully consider them.

A) According to author: "The equation for the evolution step is the same in both formalisms, it is Eq. 1 (the third component is never used in the hydrostatic formalism as the vertical eulerian-acceleration is calculated based on the horizontal divergence). I now added in the beginning of section 3:...".

   I cannot translate this sentence into a clear mathematical equation and there are still some elements that are mathematically unclear. For example, I think that my equations (R2) to (R6) written in my previous review correspond to the mathematical translation of the author's sentence. If so, then the author should specify equations (R2) - (R6), if not, then the author should write the equation that gives (19) (new version). This is an important point that I maintain, because it creates some confusion about the mathematical formulation. However, the core of this article is the quantification of the differences between two formalisms.

B) According to author: "I disagree with the reviewer $\mathbb{P}^N$ is perpendicular to $\mathbf{k}$, but $\mathbb{P}^H$ is not"

   In fact, to be precise, according to your Figure 2 (new version), $\mathbb{P}^N(\hat{\mathbf{a}})$ is the **orthogonal** projection of $\hat{\mathbf{a}}$ onto the plane perpendicular to $\mathbf{k}$ while $\mathbb{P}^H(\hat{\mathbf{a}})$ is the **vertical** (non-orthogonal) projection (non-orthogonal) projection of $\hat{\mathbf{a}}$ in the plane perpendicular to $\mathbf{k}$. These projections for the two formalisms are always in a plane perpendicular to $\mathbf{k}$ to respect the free divergence. This geometric difference between the orthogonal projection and the vertical projection reflects the central difference between the two formalisms. This distinction must be clearly indicated in the legend of Figure 2.

C) According to Author "The reviewer is right, there is a difference in the definition of $\mathbb{P}^N$ and its representation in the Figure explaining the projection, because $\mathbb{P}^H$ is not only the projection in the vertical "down" on the subspace of zero divergence (this only the last part), there are two more steps involved: the forgetting of the vertical dynamic acceleration and the addition of the buoyancy acceleration to the horizontal acceleration. The

Fig. and its legend are now corrected. In the text it was ok. When I give a talk on the subject I spent 10 min on this fig. as it explains all the projection formalism. I added in the text:..."

For a reader like me, we need to know where we are going when we start a new section. Indeed, we do not know why to extend the fourth dimension to the operator, and it takes time (several hours in my case) to finally understand the interest of the mathematical tricks. This means that the author must start explaining the issues at the beginning of the section, for example:

"The purpose of this section is to express the operator $\mathbb{A}$ that measures the difference between the two formulations $\mathbb{P}^N$ and $\mathbb{P}^H$ that are defined in (20) and (22) (new version). However, the problem comes from the difference in the input vectors $a^N$ and $a^H$ in the operators $\mathbb{P}^N$ and $\mathbb{P}^H$. In order to express a single input of the operator $\mathbb{A}$ we use a vector $\tilde{a}$ which is defined in (24) (new version) by finding a common basis between the operator $\mathbb{P}^N$ and $\mathbb{P}^H$."

D) According to author "The square-root of difference in the square of the horizontal wave velocity between the two formalisms... phase speed."
What is the definition of $c^N$ and $c^H$? It should be specified in the text (I assume they are the phase velocities)..

---

## Referee Report (RR2)

The author has satisfactorily answered my comments, and those of the other two reviewers.
The paper has improved in clarity and significance.
It can be accepted in the present form.

---

## Author Response (AR2)

Answer to the reviewers 3 second report:

Dear Editor,

I am grateful to Reviewer 3 as his criticism is to the point and continues to increase the quality of the paper. Please find my detailed answers and corrections to the reviewer's comments (reproduced in black and blue) below, written in olive. The corrections performed to the manuscript are given in red and an updated version of the manuscript with the corrections highlighted in red is provided.

Sincerely,

Achim Wirth

Report on
On the hydrostatic approximation in rotating stratified flow
by Achim Wirth
Review 2

The new version of the text presented by the author has been much improved. Nevertheless, there are still grey areas that need to be clarified. I suggest that the result of this studies would be appropriate for publication if the author revised the manuscript taking into account the comments below. I am confident that author will carefully consider them.

A) According to author: "The equation for the evolution step is the same in both formalisms, it is Eq. 1 (the third component is never used in the hydrostatic formalism as the vertical eulerian-acceleration is calculated based on the horizontal divergence). I now added in the beginning of section 3:...".

I cannot translate this sentence into a clear mathematical equation and there are still some elements that are mathematically unclear. For example, I think that my equations (R2) to (R6) written in my previous review correspond to the mathematical translation of the author's sentence. If so, then the author should specify equations (R2) - (R6), if not, then the author should write the equation that gives (19) (new version). This is an important point that I maintain, because it creates some confusion about the mathemat- ical formulation. However, the core of this article is the quantification of the differences between two formalisms.

The reviewer is right, the hydrostatic equation was missing, it is now added (his eq. (R4)) as my eq. (3). His eqs. (R2) and (R3) are my eq. (1) (in vector form) the divergence condition (R6) is my equation (2). I added his eq. (R5) during the first review, as my (now) eq. (5). In section 2 I added:

When the velocity vanishes, the above equations simplify to the hydrostatic equation:

$$\partial_z P = -g\frac{\rho}{\rho_0} = \tilde{b},\tag{1}$$

showing that the vertical pressure gradient equals the buoyancy.

And I refer to this equation in section 3: (Eq. 1)

B) According to author: "I disagree with the reviewer $\mathbb{P}^N$ is perpendicular to $\mathbf{k}$, but $\mathbb{P}^H$ is not"
In fact, to be precise, according to your Figure 2 (new version), $\mathbb{P}^N(\hat{\mathbf{a}})$ is the orthogonal projection of â onto the plane perpendicular to $\mathbf{k}$ while $\mathbb{P}^H(\hat{\mathbf{a}})$ (â) is the vertical (non- orthogonal) projection (non-orthogonal) projection of â in the plane perpendicular to $\mathbf{k}$. These projections for the two formalisms are always in a plane perpendicular to $\mathbf{k}$ to respect the free divergence. This geometric difference between the orthogonal projection and the vertical projection reflects the central difference between the two formalisms. This distinction must be clearly indicated in the legend of Figure 2.

The reviewer is right, my formulation in the answer was wrong. The right formulation is: $\mathbb{P}^N$ is along $\mathbf{k}$ and perpendicular onto the double red line. I now added to the legend of Figure 2:

it is orthogonal to the $\mathbf{k}$ vector.

And corrected:

( thick-blue-vector)

Other small improvements were done on the legend.

C) According to Author "The reviewer is right, there is a difference in the definition of $\mathbb{P}^N$ and its representation in the Figure explaining the projection, because $\mathbb{P}^H$ is not only the projection in the vertical "down" on the subspace of zero divergence (this only the last part), there are two more steps involved: the forgetting of the vertical dynamic acceler- ation and the addition of the buoyancy acceleration to the horizontal acceleration. The Fig. and its legend are now corrected. In the text it was ok. When I give a talk on the subject I spent 10 min on this fig. as it explains all the projection formalism. I added in the text:..."
For a reader like me, we need to know where we are going when we start a new section. Indeed, we do not know why to extend the fourth dimension to the operator, and it takes time (several hours in my case) to finally understand the interest of the mathematical tricks. This means that the author must start explaining the issues at the beginning of the section, for example: "The purpose of this section

is to express the operator $\mathbb{A}$ that measures the difference between the two formulations $\mathbb{P}^N$ and $\mathbb{P}^H$ that are defined in (20) and (22) (new version). However, the problem comes from the difference in the input vectors $\mathbf{a}^N$ and $\mathbf{a}^H$ in the operators $\mathbb{P}^N$ and $\mathbb{P}^H$ . In order to express a single input of the operator $\mathbb{A}$ we use a vector $\tilde{\mathbf{a}}$ which is defined in (24) (new version) by finding a common basis between the operator $\mathbb{P}^N$ and $\mathbb{P}^H$."

I now changed the first paragraph of section 4 along the lines of the Reviewers comments:

In this section I introduce the extension of the Fourier space to 4 dimension, this technique is labeled T2 in the introduction and in Fig. ??. It allows to use a common mathematical formalism, based on 4 dimensional linear algebra, for the Navier-Stokes and the hydrostatic projection. In the 4 dimensional space both projections apply to the same vector ($\tilde{a}$, introduced below) and it is then straightforward to obtain the difference between the two projections, four dimensional matrices ($\mathbb{A}$, given at the end of this section). Only the entries of the matrices differ between the two formalisms. Increasing the dimension from three to four is necessary as the two parts of the vertical acceleration, dynamical and buoyancy are treated differently between the formalisms.

D) According to author "The square-root of difference in the square of the horizontal wave velocity between the two formalisms... phase speed." What is the definition of $c^N$ and $c^H$ ? It should be specified in the text (I assume they are the phase velocities)..
I now added:

The phase velocities are given by $c^N = \omega^N/k$ and $c^H = \omega^H/k$, where the wave frequencies are defined by the dispersion relation in Eqs. ??.